# QUEEN: QUantized Efficient ENcoding of Dynamic Gaussians for Streaming Free-viewpoint Videos

**Sharath Girish**[*]
University of Maryland
sgirish@cs.umd.edu

**Tianye Li**[†]
NVIDIA
tianyel@nvidia.com

**Amrita Mazumdar**[†]
NVIDIA
amritam@nvidia.com

**Abhinav Shrivastava**
University of Maryland
abhinav@cs.umd.edu

**David Luebke**
NVIDIA
dluebke@nvidia.com

**Shalini De Mello**
NVIDIA
shalinig@nvidia.com

## Abstract

Online free-viewpoint video (FVV) streaming is a challenging problem, which is relatively under-explored. It requires incremental on-the-fly updates to a volumetric representation, fast training and rendering to satisfy real-time constraints and a small memory footprint for efficient transmission. If achieved, it can enhance user experience by enabling novel applications, *e.g.*, 3D video conferencing and live volumetric video broadcast, among others. In this work, we propose a novel framework for QUantized and Efficient ENcoding (QUEEN) for streaming FVV using 3D Gaussian Splatting (3D-GS). QUEEN directly learns Gaussian attribute residuals between consecutive frames at each time-step without imposing any structural constraints on them, allowing for high quality reconstruction and generalizability. To efficiently store the residuals, we further propose a quantization-sparsity framework, which contains a learned latent-decoder for effectively quantizing attribute residuals other than Gaussian positions and a learned gating module to sparsify position residuals. We propose to use the Gaussian viewspace gradient difference vector as a signal to separate the static and dynamic content of the scene. It acts as a guide for effective sparsity learning and speeds up training. On diverse FVV benchmarks, QUEEN outperforms the state-of-the-art online FVV methods on all metrics. Notably, for several highly dynamic scenes, it reduces the model size to just 0.7 MB per frame while training in under 5 sec and rendering at ~350 FPS.

## 1 Introduction

The dynamic world that we perceive around us is not 2D, but rather 3D. Unlike 2D videos, which are ubiquitous, the question of how to effectively capture, encode and disseminate free-viewpoint videos (FVV) of dynamic 3D scenes, which can be viewed at any instance of time and from any viewpoint, has intrigued computer vision and graphics researchers for much time. Free-viewpoint video transmission, if achieved, has the potential to transform and enrich user experience in profound ways by offering novel immersive experiences, *e.g.*, FVV video playback and live streaming, 3D video conferencing and telepresence, gaming, virtual spatial tutoring and teleoperation, among others.

The underlying problem of reconstructing FVV involves learning a 6D plenoptic function of a dynamic scene $P(\boldsymbol{x}, \boldsymbol{d}, t)$ from sparse multiple views acquired over a window of time, with $\boldsymbol{x} \in \mathbb{R}^3$ being a position in 3D space, $\boldsymbol{d} = (\theta, \phi)$ a viewing direction and $t$ an instance of time. Neural volumetric representations, which learn a 5D plenoptic function of a scene $P(\boldsymbol{x}, \boldsymbol{d})$ at a fixed time instance, *e.g.*,

---

[*]SG's work was done during an internship at NVIDIA. [†]TL and AM contributed equally to this project.
Project website: https://research.nvidia.com/labs/amri/projects/queen.

38th Conference on Neural Information Processing Systems (NeurIPS 2024).

neural radiance fields (NeRFs) [54] and its variants [2, 7, 18, 55] present a compact and high-fidelity representation for 3D scenes. NeRFs have also been extended to dynamic 4D scenes [1, 5, 19] providing a powerful tool for reconstructing FVV. However, NeRFs require compositing dense information across a 3D volume and hence are slow to train and render. Recently, 3D Gaussian Splatting (3D-GS) [29] has emerged as a promising technique with significantly faster training and rendering speeds in comparison with NeRFs, and they have also been extended to dynamic 4D scenes [42, 83, 88]. While these representations accurately model 4D scenes, they are trained in an *offline* fashion requiring full multi-view video sequences to learn temporal relationships between frames. They also require long training times to achieve high reconstruction quality and are mostly not streamable.

*Online* FVV, *e.g.*, for broadcast and teleconferencing applications, presents additional challenges versus offline. It requires incremental *on-the-fly* updates to volumetric representation at each time-step of the dynamic scene, *fast* training and rendering times to maintain real-time operation, and *small* packet sizes per frame to enable effective transmission on bandwidth-limited channels. Consequently, the more challenging problem of online FVV reconstruction remains relatively under-explored. Notable prior solutions are those based on NeRFs using voxel grids [37, 78] or triplanes [84] to learn 3D representations that are updated on-the-fly. Unsurprisingly, they suffer from slow rendering speeds. Recently, Sun et al. proposed 3DGStream [69], which uses 3D-GS to model a 3D scene along with InstantNGP [55] to model its geometric transformation over time. It achieves high rendering speeds but imposes heuristic structural constraints on the volumetric representation to achieve efficiency, which compromises model expressiveness and quality.

In this work, we propose a novel QUantized and Efficient ENcoding (QUEEN) framework, which uses 3D-GS for online FVV. Similarly to prior approaches [69], we also learn Gaussian attribute residuals between consecutive time-steps. To reduce memory requirements, however, [69] learns only a subset of the Gaussian attributes at each time-step, limiting model expressiveness. Our first insight, therefore, is to model residuals for *all* attributes instead, which does not compromise quality. However, encoding all Guassian attributes increases the per frame memory requirement and hence necessitates a means to compress them more effectively. Our second insight, then, is to learn to directly compress the Gaussian residuals in proportion to the real-time scene dynamics, *e.g.*, motion and illumination changes. This contrasts with existing methods [37, 69, 78, 84] that employ a single fixed-sized structure, *e.g.*, a voxel-grid, a triplane, or hash encoding at all time-steps, and the result is higher efficiency in terms of model size, training speeds, and rendering speeds. Lastly, we also exploit temporal redundancies across time-steps to limit computations to the highly dynamic parts of the scene only and achieve further efficiencies.

Specifically, to achieve this, we propose a learned quantization-sparsity framework to simultaneously learn and compress Gaussian attribute residuals for each time-step. We quantize all attribute residuals, except Gaussian positions, via an end-to-end trainable integer-based latent-decoder. Once learned, we efficiently encode the integer latents via entropy coding to achieve high compression factors. For position residuals that exhibit greater sensitivity to quantization, we propose a learned gating mechanism to sparsify them, which identifies the static (corresponding to $0$ value) and dynamic Gaussians and retains the sparse dynamic ones only at full precision. Finally, to achieve further efficiencies in terms of training time and storage, we utilize the differences between the 2D viewspace Gaussian gradients of consecutive frames to initialize our learnable gates, and to selectively render local image regions corresponding to highly dynamic scene content.

We evaluate our approach, QUEEN, on two benchmark datasets, containing diverse scenes with large geometric motion and illumination changes. QUEEN outperforms all prior state-of-the-art approaches for online FVV and significantly reduces the per-frame memory cost ($\sim$10$\times$), all while achieving higher reconstruction quality, as well as faster training and rendering speeds. Extensive ablations show the efficacy of the various components of our approach.

To summarize, our key contributions are:

- We propose a Gaussian residual-based framework to model 3D dynamic scenes for online FVV without any structural constraints, which allows free learning of all 3D-GS attribute residuals, resulting in higher model expressiveness.

- We introduce a learned quantization-sparsity framework for compressing per-frame residuals, and we initialize and train it efficiently using viewspace gradient differences that separate the dynamic and static scene content.

- On various challenging real-world dynamic scenes, we surpass existing state-of-the-art approaches on all metrics: reconstruction quality, memory utilization, as well as training and rendering speed.

## 2 Related Work

### 2.1 Traditional Free-viewpoint Video

Ever since early FVV work such as [27], a series of geometry-based FVV methods [12, 57] has been pushing for high reconstruction quality and streamable performance. However, their rendering and compression quality rely on the accuracy of a sophisticated pipeline of geometry reconstruction [20, 28], tracking [36], and texturing [61]. They also require high-end hardware for capturing complex and dynamic appearance [6, 14, 26]. Purely image-based rendering [10, 13, 35, 53] relaxes the requirement for geometric accuracy. Although methods such as [4, 95] support view interpolation with layered representations in the dynamic setting, they require a high count of views as input to ensure interpolation quality.

### 2.2 Neural and Gaussian-based Free-viewpoint Video

**Offline Methods.** Compared to the traditional representations, the emergence of neural representations [47, 54, 86, 67, 72] opened a new door for capturing FVV for dynamic humans [21, 31, 32, 39, 63, 82, 94] and monocular videos [22, 43, 44, 73, 75, 85]. In this work, we focus on general dynamic scenes [92] from multiple views to push the quality of streamable FVV *without* requiring a strong human prior [40, 48] or a very constrained input. [16, 62, 74, 80] model the scene dynamics via explicit deformation. Although suitable for motion analysis, they inevitably face a trade-off between motion accuracy and visual quality [74]. To tackle this, [41, 52, 59] use a spatial-temporal formulation via time-conditioned latent codes to implicitly encode the 4D scene, enabling reconstruction of topological changes and volumetric effects. [5, 19, 66] factorize the 4D scene into multiple space-time feature planes and achieves higher model compactness and training efficiency. [68, 76] decompose the 4D scene into static and dynamic volumes. [1, 46, 77, 87, 93] incorporate efficient NeRF representations [9, 19, 91] for higher fidelity. Although, these NeRF-based method achieve high compactness, they suffer from low rendering efficiency, even when converted [68] to a more efficient NeRF formulation [9, 55]. Seeing their great potential for efficiency, recent works extend 3D Gaussian representations [29] to dynamic scenes, with temporal attributes [42, 89], generalized 4D Gaussians [88] and a hybrid representation [83]. While these methods achieve high quality in modeling 3D dynamic scenes, they, together with the aforementioned NeRF-based methods, are mostly *offline*, i.e., they require all the input video and a long time for training, which is inherently difficult for streaming applications.

**Online Methods.** *Online* reconstruction for FVVs is relatively under-explored, as it imposes additional challenges of on-the-fly reconstruction using only local temporal information instead of the full recordings. Furthermore, toward the goal of streamable FVVs, the encoding system is evaluated by multiple metrics including compression rate, encoding and rendering speed and visual quality. [50] tracks dense 3D Gaussians by solving their motion over time. Visual quality and dynamic appearance is not their focus. [23] models motion by rendering scene dynamics, however their method is not optimized for efficiency. [45] focuses on generalizable NeRF reconstruction and shows good promise to adapt to a new frame but has a high memory footprint due to an MVSNet-style neural network [8, 90]. [37] accelerates training and rendering speed with a special tuning strategy and sparse voxels, however, their representation still has high temporal redundancy. [81] proposes an incremental training scheme with natural information partitioning and achieves high compression, but its encoding is slow. Several works [78, 79, 84] use video codec-inspired encoding paradigms for data efficiency. [78] achieves a decent compression rate and near interactive rendering with compact motion and residual grids. However, their training requires 10 minutes per frame. [79] focuses on real-time decoding, streaming and rendering instead of on-the-fly encoding. [84] performs grouped training on a hybrid representation of triplanes and volume grid. While achieving high compression rate, their fixed encoding paradigm and aggressive quantization limits their reconstruction quality along-with low rendering speeds. [69] is the closest work to ours for streaming FVV via 3D-GS. They encode the position and rotation residuals via an Instant-NGP [55] based transformation cache. While achieving faster training and rendering speeds than prior work, they have high data redundancy due to a fixed structured modeling. Additionally, they focus on geometric transformations only and

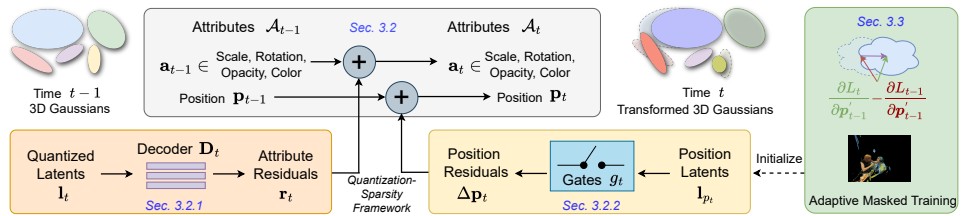

Figure 1: **Overview of QUEEN for online FVV.** We incrementally update Gaussian attributes at each time-step (gray block) by simultaneously learning and compressing residuals between consecutive time-steps via a quantization (orange block) and sparsity (yellow block) framework. We additionally render only the dynamic Gaussians for masked regions to achieve faster convergence (green block).

can approximate only small changes in new scene content or lighting variations. Our work updates and compresses all 3D-GS attributes freely without any structural constraints while still obtaining much better memory costs via our quantization-sparsity framework along with better training times.

### 2.3 3D Scene Representation Compression

Several works propose a variety of compression methodologies for reducing the memory, training time, or rendering speed of standard static scene 3D representations. [9, 71] decompose NeRFs via low-rank approximations. [15, 38] prune voxels along with vector quantization by [38]. [25, 70] compress multi-resolution feature-grids via codebook/vector quantization. A large number of approaches target 3D-GS compression [17, 24, 34, 58] and acceleration via pruning. While these approaches can be applied to static representations on a per-frame basis, their trivial frame-wise application would result in extremely high training costs as well as large memory per frame. To enable streaming, our work, instead, explicitly focuses on effectively leveraging the temporal redundancies across frames by compressing the residual information between them to achieve greater efficiency.

## 3 QUEEN: Quantized Efficient Encoding for Streaming FVV

A solution for streamable FVV must have low-latency encoding (training) and decoding (rendering), and low data bandwidth (memory) for transmission on a common network infrastructure. Motivated by these constraints, we aim to generate streamable FVVs with compact representations that are fast to train and render incrementally. In this section, we first provide an overview of 3D-GS (Sec. 3.1). In Sec. 3.2, we propose a compression framework to efficiently represent and train Gaussian attribute residuals at each time step. Sec. 3.3 discusses utilizing an approach based on viewspace gradient differences to achieve greater efficiencies. An overview of our method is shown in Fig. 1.

### 3.1 Preliminary: 3D Gaussian Splatting

Our efficient representation for dynamic scenes is based on 3D Gaussian Splatting (3DGS) [29]. Given multi-view images $\mathcal{I}$, a 3D scene is modeled by a set of Gaussians with attributes $\mathcal{A}$.

**Representation.** The shape of each Gaussian $i$ is defined by its mean $\mathbf{p}_i \in \mathbb{R}^3$ and covariance matrix $\mathbf{\Sigma}_i$. The covariance matrix is represented by $\mathbf{\Sigma}_i = \mathbf{R}_i \mathbf{S}_i \mathbf{S}_i^T \mathbf{R}_i^T$, where $\mathbf{R}_i$ is a rotation matrix parameterized by a quaternion vector $\mathbf{q}_i \in \mathbb{R}^4$, and the scale matrix $\mathbf{S}_i$ is a diagonal matrix with elements $\mathbf{s}_i \in \mathbb{R}^3$. Each Gaussian also contains opacity $o_i \in [0, 1]$ and spherical harmonic coefficients $\mathbf{h}_i$ for view-dependent appearance with dimensions based on the number of degrees.

**Rendering.** For rasterization, 3D Gaussians are projected into 2D Gaussians for any given view. Given a camera with intrinsic matrix $\mathbf{K}$ and viewing transform $\mathbf{W}$, the 2D mean and covariance are:

$$\mathbf{p}_i' = \Pi(\mathbf{p}_i; \mathbf{K}, \mathbf{W}), \quad \mathbf{\Sigma}_i' = \mathbf{J} \mathbf{W} \mathbf{\Sigma}_i \mathbf{W}^T \mathbf{J}^T, \tag{1}$$

where $\Pi(\cdot)$ denotes the perspective projection and $\mathbf{J}$ is the Jacobian of the affine approximation of the projective transform [96]. The image color $\hat{\mathbf{c}}$ at pixel location $\mathbf{x}$ is obtained by blending $N$ depth-sorted Gaussians with their view-dependent RGB color value $\mathbf{c}_i$ computed from $\mathbf{h}_i$:

$$\hat{\mathbf{c}}(\mathbf{x}) = \sum_{i=1}^{N} \mathbf{c}_i \alpha_i \prod_{j=1}^{i-1}(1 - \alpha_j), \quad \alpha_i = o_i \cdot \exp\left(-\frac{1}{2}(\mathbf{x} - \mathbf{p}_i')^T \mathbf{\Sigma}_i'^{-1}(\mathbf{x} - \mathbf{p}_i')\right), \tag{2}$$

where $\alpha_i$ is the conic opacity of Gaussian $i$ at pixel location $\mathbf{x}$ multiplied by the Gaussian opacity $o_i$.

**Training.** With a differentiable rasterizer [29], the attributes $\mathcal{A} = \{\mathbf{p}_i, \mathbf{q}_i, \mathbf{s}_i, o_i, \mathbf{h}_i\}_{i=1}^{N}$ are optimized to produce renderings $\hat{\mathcal{I}} = R(\mathcal{A})$ that fit to input images $\mathcal{I}$ by optimizing a reconstruction loss which combines the D-SSIM loss [65] and $L_1$ loss with a hyperparameter $\lambda$:

$$L = \lambda L_{\text{D-SSIM}} + (1 - \lambda)L_1. \tag{3}$$

### 3.2 Attribute Residual Compression

Given a multi-view image sequence $\{\mathcal{I}_t\}_{t=0}^{T-1}$, our goal is to reconstruct the dynamic scene via Gaussian attributes $\mathcal{A}_t$ for each time-step $t$. We model the attributes based on the trained attributes from the previous time-step $t - 1$ as

$$\mathcal{A}_t = \mathcal{A}_{t-1} + \mathcal{R}_t, \tag{4}$$

where $\mathcal{R}_t$ consists of learnable residuals for each attribute (in Fig. 1 (gray block)). For time-step $t = 0$, we perform vanilla Gaussian splatting training to obtain attributes $\mathcal{A}_0$. This sequential formulation allows us to freely and adaptively update the residuals $\mathcal{R}_t$ on-the-fly with incoming streaming training views, without any structural constraints as in prior works [69]. However, representing the 4D scene with uncompressed residuals is still highly inefficient. As residuals have low magnitudes in comparison with the attributes themselves, they can be efficiently compressed, for which we propose a novel quantization-sparsity framework.

#### 3.2.1 Attribute Residual Quantization

There exists spatial redundancy within the Gaussian attributes of the same time-step. Nearby Gaussians have highly correlated residuals for shape, orientation and appearance. To reduce the storage cost of the residuals, we propose to utilize a quantization framework during training [24].

At each time-step $t$, we represent the residuals via quantized latents and a shared compact decoder. Specifically, to obtain the residuals for each category[2] $\mathbf{r}_i \in \mathbb{R}^M$, we maintain corresponding quantized integer latents $\mathbf{l}_i \in \mathbb{Z}^L$ for each Gaussian $i$. These latents are passed through a shared linear decoder $D$ with learnable parameters $\mathbf{D} \in \mathbb{R}^{M \times L}$ to obtain the decoded attribute residual $\mathbf{r}_i$. Such a compact decoder has small time and memory costs due to few parameters and arithmetic operations. To allow differentiable training of the integer latents via gradient optimization, we use a continuous approximation $\hat{\mathbf{l}}_i \in \mathbb{R}^L$ instead. $\hat{\mathbf{l}}_i$ are rounded to the nearest integer values for the forward pass but can still receive backpropagated gradients via the Straight-Through Estimator (STE) [3]:

$$\mathbf{l}_i = \text{STE}(\hat{\mathbf{l}}_i), \quad \mathbf{r}_i = D(\mathbf{l}_i; \mathbf{D}) = \mathbf{D} \cdot \text{float}(\mathbf{l}_i). \tag{5}$$

The continuous latents $\hat{\mathbf{L}} = \{\hat{\mathbf{l}}_i\}_{i=1}^{N}$, and the shared decoder's parameters $\mathbf{D}$ are learnable during training. After adding the decoded residuals to the previous time-step's attributes (Eq. 4), the standard rasterization process (Eq. 2) is used to obtain the rendered image. This differentiable quantization module is trained end-to-end with the main training process by optimizing the reconstruction loss. Post-training, we entropy code the quantized latents $\mathbf{L}$ and directly store the decoder $\mathbf{D}$. Entropy coding results in as much as $10\times$ reduction in model size from $44$ to $4$ MB without quality degradation.

#### 3.2.2 Position Residual Gating

**Sparse Representation.** While most of the attribute residuals can be quantized effectively with our proposed method in Sec 3.2.1, we observe that the position residuals are sensitive to quantization and require high precision during rendering[3]. Storing all the full-precision position residuals, however, still results in high per-frame memory costs. To tackle this, we propose a learned gating methodology, which enforces sparsity in the residuals instead of quantization. This mechanism allows us to set a vast majority of the position residuals to zeros, while maintaining full-precision non-zero values. Specifically, we represent the positional residual for each Gaussian $i$ as $\Delta\mathbf{p}_i = g_i \cdot \mathbf{l}_{p_i}$, where the scalar $g_i$ is the learnable gate variable and $\mathbf{l}_{p_i} \in \mathbb{R}^3$ is the learnable pre-gated residual in full precision during training. After training, the sparse $\Delta\mathbf{p}_i$ can be efficiently stored via sparse matrix formats [60] to reduce memory costs. Thus, our goal is to encourage the sparsity for the variable $g_i$ across all

---

[2]$\mathbf{r} \in \mathcal{R}$, belong to one of five categories of Gaussian attributes: position, rotation, scale, opacity and color.
[3]Concurrent work [58] also discovered that positional attributes are sensitive to compression.

Gaussians. This goal also aligns with the observation that a large portion of a dynamic scene is static or nearly static, which can be leveraged to attain high compression performance.

**Hard Concrete Gate.** Sparsity can be induced via $L_0$ or $L_1$ norm regularization penalties. However, $L_1$ norm induces shrinkage, *i.e.*, lowers the magnitude of even non-zero values. $L_0$ norm is the ideal sparsity loss without shrinkage, but is computationally intractable with non-differentiability and combinatorial complexity. To enforce sparsity, we instead propose to use the hard concrete gate [49]. For each Gaussian $i$, the concrete gate [51] is a continuous relaxation of the Bernoulli distribution:

$$\hat{g}_i = \text{Sigmoid}(\log \alpha_i / \tau), \tag{6}$$

where $\alpha_i$ is a learnable parameter and $\tau$ is the temperature parameter. Although the concrete gate approximates the discrete Bernoulli gate, it does not include the end points $\{0, 1\}$, which does not directly result in sparsity. The hard concrete gate "stretches" the range of the concrete gate to the interval $(\gamma_0, \gamma_1)$ and then applies a hard-sigmoid:

$$\widetilde{g}_i = \hat{g}_i \cdot (\gamma_1 - \gamma_0) + \gamma_0, \quad g_i = \min(1, \max(0, \widetilde{g}_i)). \tag{7}$$

This includes the end points $\{0, 1\}$ for $g_i$, required for achieving sparse residuals.

**Sparsity Loss.** The hard concrete gate formulation leads to a convenient regularization loss for encouraging $L_0$ sparsity [49] in the gates $g_i$:

$$L_{\text{reg}} = \sum_{i=1}^{N} p_i = \sum_{i=1}^{N} \text{Sigmoid}\left(\log \alpha_i - \tau \log \frac{-\gamma_0}{\gamma_1}\right). \tag{8}$$

Here, we treat $\boldsymbol{\alpha} = \{\alpha_i\}_{i=1}^{N}$ to be learnable parameters for all $N$ Gaussians at a given time-step and $\{\tau, \gamma_0, \gamma_1\}$ as hyperparameters that are shared for all Gaussians and all time-steps.

### 3.3 Viewspace Gradient Difference for Adaptive Training

Real-world dynamic scenes contain high amounts of temporal redundancy with only a fraction of the content changing between consecutive time-steps. The proposed quantization-sparsity framework can learn to identify Gaussians corresponding to static scene content and set their residuals to 0. However, they still forward/backward pass through static regions resulting in wasted training computation. Additionally, initializing the gates with 1s requires more iterations for convergence. We thus propose a proxy metric to identify Gaussians, which are static or dynamic at the start of training. We use this metric to initialize our gates while also identifying dynamic image regions to perform local rendering in, during training.

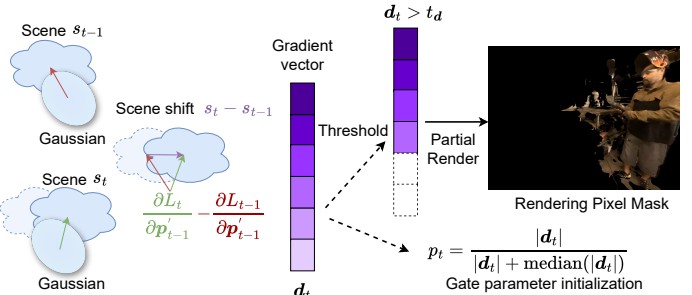

Figure 2: **Viewspace Gradient Difference.** We use the difference of viewspace gradients between consecutive frames to identify dynamic scene content.

**Viewspace Gradient Difference.** The ground-truth (GT) training images contain information of the dynamic scene content, which we leverage to separate static and dynamic Gaussians. A simple pixel difference between consecutive frames does not account for illumination changes and is a noisy signal for the geometric position residuals. 3D-GS utilizes 2D viewspace gradients $\frac{\partial L}{\partial \mathbf{p}'}$ to identify poorly fitted Gaussians based on the reconstruction loss $L$ between the rendered ($\hat{\mathcal{I}}$), GT image ($\mathcal{I}$).

More concretely, after training Gaussians at time-step $t-1$ with an MSE loss, $L_{t-1}$ and 2D Gaussian means $\mathbf{p}'_{t-1}$, we compute the MSE loss for the next time-step $L_t$ and then compute the gradient difference. The score vector $\mathbf{d}_t$ is the average of gradient differences across all training views $v$:

$$\mathbf{d}_t = \frac{1}{V} \sum_{v=0}^{V-1} \left[ \frac{\partial L_t^{(v)}}{\partial \boldsymbol{p'}_{t-1}^{(v)}} - \frac{\partial L_{t-1}^{(v)}}{\partial \boldsymbol{p'}_{t-1}^{(v)}} \right], \quad L_{t-1}^{(v)} = L\left(\mathcal{I}_{t-1}^{(v)}, \hat{\mathcal{I}}_{t-1}^{(v)}\right), \quad L_t^{(v)} = L\left(\mathcal{I}_t^{(v)}, \hat{\mathcal{I}}_{t-1}^{(v)}\right). \tag{9}$$

As shown in Fig. 2, $\boldsymbol{d}_t$ identifies the dynamic scene regions while factoring out the noise from imperfect reconstructions at time-step $t-1$. We use the norm of the score vector $|\mathbf{d}_{t_i}|$ to initialize the gate parameters. We define the probability of a gate being active for Gaussian $i$ at time-step $t$ as

$$d_{t_i} = \frac{|\mathbf{d}_{t_i}|}{|\mathbf{d}_{t_i}| + \underset{i=1,2,...,N}{\text{median}}(|\mathbf{d}_{t_i}|)}. \tag{10}$$

We set $p_i$ in Eq. 8 to be $d_{t_i}$ to solve for the initial $\alpha_i$. This initialization leads to better convergence by identifying Gaussians corresponding to 0 position residuals at the start of training itself.

**Adaptive Masked Training.** In addition to gate initialization, we propose to utilize the score vector $\boldsymbol{d}_t$ for an adaptive masked training scheme. We split the Gaussians into static or dynamic parts by applying a threshold $t_{\mathbf{d}}$ on the norm of $\boldsymbol{d}_t$. We render dynamic Gaussians for each training view to identify corresponding dynamic image regions. We then only render and backpropagate through these pixel locations. We perform this masked training for a fraction of the full training iterations, and find it to improve training speeds with little to no loss of reconstruction quality.

### 3.4 Efficient End-to-end Learnable Residuals

**Initial Frame Reconstruction.** For an incrementally updating online approach, it is important to make sure the initial frame is well reconstructed. COLMAP used to initialize the positions of Gaussians can result in sparse 3D points for regions with sparse camera views. Hence, we use an off-the-shelf monocular depth estimation network to estimate point locations in these empty regions and predict a more complete initial point cloud. Further details and results are in the supplementary.

**End-to-end Training.** We train separate decoders and quantized latents $\{\mathbf{L}_c, \mathbf{D}_c | c \in \{\boldsymbol{q}, \boldsymbol{s}, o, \boldsymbol{h}\}\}$ for all attributes except position. For position, we learn the gate parameters $\boldsymbol{\alpha}$ and positional residuals $\mathbf{L}_p$. All variables are end-to-end differentiable. The total loss function that we minimize is the reconstruction loss (Eq. 3) and the sparsity gate regularization loss (Eq. 8):

$$L_{\text{total}} = L + \lambda_{\text{reg}} L_{\text{reg}}, \tag{11}$$

where $\lambda_{\text{reg}}$ controls tradeoffs between memory and reconstruction quality. By simultaneously quantizing while training we achieve high compression while maintaining quality, unlike [84, 78] with post-training compression that lead to quality degradations. We also apply the 3D-GS densification stage at each time-step and is sufficient in modeling new or finer scene content. 3DGStream [69] adds Gaussians relative to the first time-step only, which limits their approach to small scene changes.

## 4 Experiments

### 4.1 Datasets and Implementation

We evaluate our method on two challenging FVV video datasets. **(1) Neural 3D Videos (N3DV)** [41] consists of six indoor scenes with forward-facing 20-view videos. **(2) Immersive Videos** [4] consists of seven indoor and outdoor scenes captures with 46 cameras. In both datasets, the central view is held out for testing. We implement QUEEN on [29]. We train for 500 and 350 epochs for the first time-step, and for 10 and 15 epochs for the subsequent time-steps, for N3DV and Immersive, respectively, on an NVIDIA A100 GPU. One epoch contains all training views. We evaluate visual quality in terms of average frame-wise PSNR, SSIM, and LPIPS (VGG) across all videos. We also compute the average storage size and training time for each time-step, and the rendering speed. Additional details are provided in the supplementary materials.

### 4.2 Quantitative Comparisons

We compare QUEEN against state-of-the-art existing online FVV methods (3DGStream [69], StreamRF [37] and TeTriRF [84]) on N3DV and Immersive (Tab. 1). 3DGStream [69] is the overall best-performing prior method. Since 3DGStream [69] was originally run on an older NVIDIA V100 GPU on N3DV, we re-run 3DGStream on an NVIDIA A100 GPU on both N3DV and Immersive and denote it as 3DGStream* in Tab. 1 for consistency with QUEEN. For brevity, in Tab. 1 we additionally compare against only selected top-performing offline FVV methods. We include a more extensive comparison to all existing offline FVV methods in the supplementary (Tab. 8). Lastly, we evaluate three variants of QUEEN: QUEEN-s (small), QUEEN-m (medium) and QUEEN-l (large), with residuals trained for 6, 8 and 10 epochs, respectively.

Table 1: **Quantitative Results.** We compare QUEEN against state-of-the-art online and (a few for brevity) offline FVV methods on N3DV [41] and Immersive [4]. We include many more offline methods in the supplementary (Tab. 8). 3DGStream* refers to our re-implementation on the same NVIDIA A100 GPU used by QUEEN for fairness. Bold and underlined numbers indicate the **best** and the second best results, respectively, within each category.

Neural 3D Video (N3DV) dataset

| Category | Method | PSNR (dB) ↑ | SSIM ↑ | LPIPS ↓ | Storage (MB) ↓ | Training (sec) ↓ | Rendering (FPS) ↑ |
|---|---|---|---|---|---|---|---|
| Offline | NeRFPlayer [68] | 30.69 | 0.932 | 0.209 | 17.10 | 72 | 0.05 |
| | HyperReel [1] | 31.10 | 0.928 | - | 1.20 | 104 | 2.00 |
| | SpaceTime [42] | **32.05** | **0.948** | - | **0.67** | **20** | **140** |
| Online | StreamRF [37] | 30.68 | - | - | 31.4 | 15 | 8.3 |
| | TeTriRF [84] | 30.43 | 0.906 | 0.248 | **0.06** | 39 | 4 |
| | 3DGStream [69] | 31.67 | - | - | 7.83 | 12 | 215 |
| | 3DGStream* [69] | 31.58 | 0.941 | 0.140 | 7.80 | 8.5 | 261 |
| | QUEEN-s (ours) | 31.89 | 0.945 | 0.139 | 0.68 | **4.65** | **345** |
| | QUEEN-m (ours) | 32.03 | 0.946 | 0.137 | 0.69 | 5.96 | 321 |
| | QUEEN-l (ours) | **32.19** | 0.946 | **0.136** | 0.75 | 7.9 | 248 |

Immersive dataset

| Category | Method | PSNR (dB) ↑ | SSIM ↑ | LPIPS ↓ | Storage (MB) ↓ | Training (sec) ↓ | Rendering (FPS) ↑ |
|---|---|---|---|---|---|---|---|
| Offline | NeRFPlayer [68] | 25.8 | 0.848 | 0.329 | 17.1 | ∼72 | 0.12 |
| | HyperReel [1] | 28.8 | 0.874 | - | 1.2 | ∼108 | 4 |
| | SpaceTime [42] | **29.2** | **0.916** | - | **1.2** | ∼72 | **99** |
| Online | 3DGStream* [69] | 25.18 | 0.876 | 0.255 | 8.83 | 32.4 | **221** |
| | QUEEN-l (ours) | **29.22** | **0.915** | **0.208** | **1.79** | **19.7** | 183 |

Table 2: **Effect of Various Components Ablated on N3DV and Immersive Datasets.**

| Components | N3DV | | | | Immersive | | | |
|---|---|---|---|---|---|---|---|---|
| | PSNR (dB) ↑ | Storage (MB) ↓ | Training (sec) ↓ | Rendering (FPS) ↑ | PSNR (dB) ↑ | Storage (MB) ↓ | Training (sec) ↓ | Rendering (FPS) ↑ |
| Baseline | 31.66 | 44.36 | 7.29 | 214 | 28.54 | 78.4 | 20.85 | **276** |
| + Attribute Quantization | 32.04 | 4.18 | **7.28** | **285** | 29.01 | 4.57 | 25.17 | 199 |
| + Position Gating | 32.05 | 0.72 | 6.95 | 274 | 28.99 | **2.01** | 26.99 | 190 |
| + Gate Initialization | 32.14 | **0.60** | 7.92 | 271 | 29.08 | **1.33** | 27.81 | 177 |
| + Masked Training | **32.19** | 0.74 | 7.88 | 248 | **29.22** | 1.79 | **19.70** | 183 |

From Tab. 1, on N3DV, QUEEN-l results in the best quality among all online FVV methods and achieves a $10\times$ reduction in storage size compared to 3DGStream. Although TeTriRF requires less memory than QUEEN, it has much worse quality ($-1.5$dB) and rendering speed (4FPS), and higher training time (39 sec). On Immersive, which contains more pronounced scene changes than N3DV, we limit our comparisons to 3DGStream with longer iterations. TeTriRF requires long convergence times to achieve reasonable reconstruction quality, limiting their training feasibility. QUEEN-l significantly outperforms 3DGStream, obtaining $+4$dB PSNR, $5\times$ smaller size, and lower training times. These results on the more challenging scenes from the Immersive datasets reveal the structural constraints brought by the heuristic compression design of 3DGStream. In contrast, our quantization-sparsity framework shows higher flexibility and quality in capturing changing appearances and scene density as well as learning compact and effective representations.

### 4.3 Qualitative Comparisons

In Fig. 3 we compare the reconstruction results of the various methods. On N3DV, we reconstruct finer details than 3DGStream, *e.g.*, the hand and the dog, and minimize artifacts such as the tongs in the top scene or the coffee and metal tumbler in the bottom scene. TeTriRF produces blurry outputs, *e.g.*, the cap or metal tumbler in the bottom scene. On Immersive, we better model illumination changes and new scene content such as the person (first patch) and the flame (third patch) in the top scene or the face in the bottom scene (second patch) versus 3DGStream.

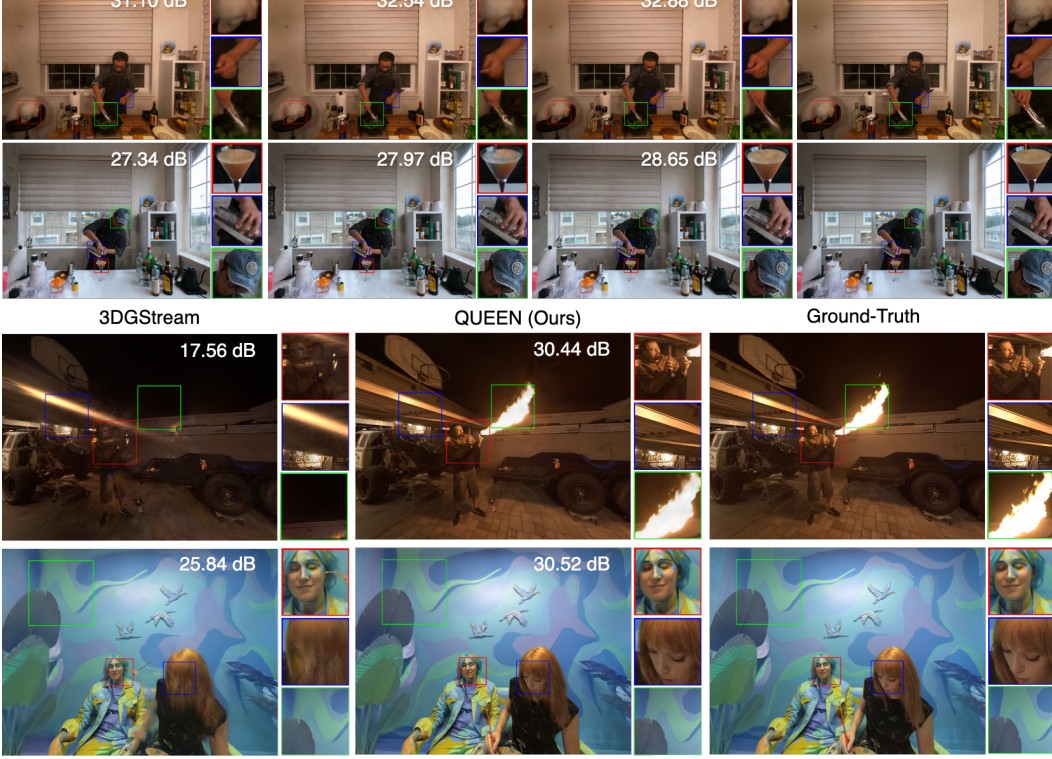

Figure 3: **Qualitative Results.** A visualization of various scenes in the N3DV and Immersive datasets. PSNR (↑) values are shown. We include additional video results in the supplement.

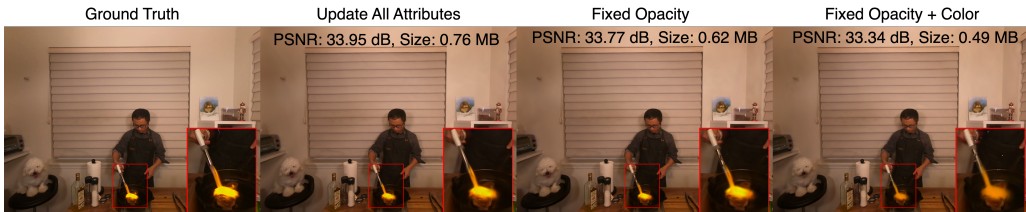

Figure 4: **Effect of Updating Appearance Attributes.** QUEEN updates all Gaussian attributes, resulting in improved quality versus keeping appearance attributes fixed across a video.

## 4.4 Ablations

**Effect of Updating Appearance Attributes.** To ablate the importance of updating the Gaussian appearance attributes (color and opacity) per frame in QUEEN, we run experiments on N3DV for 2 settings: (1) learning only geometric attribute residuals (position, scale and covariance) with appearance residuals set to zero and (2) learning all residuals per frame (Tab. 3). Updating only geometric attributes results in a drop of $0.4$ dB PSNR versus updating all attributes. Visually, for the Flame Steak scene in N3DV (Fig. 4), updating all attributes results in the highest quality, while fixing opacity introduces artifacts at the edge of the flamethrower. Fixing color, additionally, results in a significant drop in PSNR (-0.6 dB) producing a discolored flame (rightmost column).

**Effect of Attribute Compression and Masked Training.** We show results for five variants of QUEEN with incrementally added sub-components: (1) a baseline with uncompressed residual training (Sec. 3.2), (2) adding quantization to all attributes except position (Sec. 3.2.1), (3) adding gating of position residuals (Sec. 3.2.2), (4) gate initialization with viewspace gradient differences and (5) masked image training (Sec. 3.3). Results are summarized in Tab. 2 for both N3DV and Immersive datasets. Compressing attributes and gating position residuals results in significant model size reduction on both datasets $(60\times, 40\times)$. This is further reduced by gate initialization with

Table 3: **Updating Appearance Attributes on N3DV.** PSNR significantly improves by updating all attributes but with a small storage overhead.

| Update Attributes | PSNR (dB) | Storage (MB) | Train. (sec) |
|---|---|---|---|
| Geometric | 31.61 | **0.61** | 7.87 |
| + Appearance | **32.03** | 0.74 | **7.57** |

Table 4: **Effect of Quantizing Scaling Attribute on N3DV.** PSNR improves while also reducing model size and training time due to faster rendering.

| Configuration | PSNR (dB) | Storage (MB) | Train. (sec) |
|---|---|---|---|
| w/o Scaling Quant. | 31.69 | 4.39 | 11.01 |
| w/ Scaling Quant. | **32.08** | **0.69** | **7.07** |

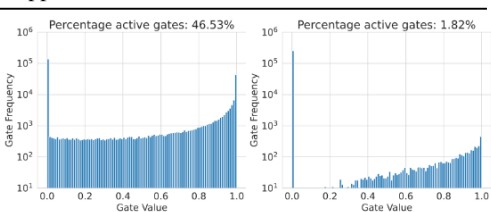 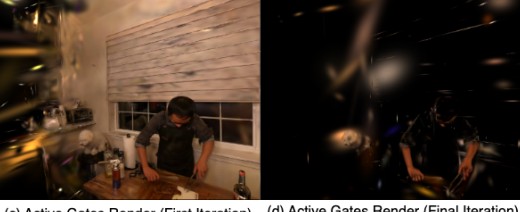

(a) Gate Histogram (First Iteration)  (b) Gate Histogram (Final Iteration)  (c) Active Gates Render (First Iteration)  (d) Active Gates Render (Final Iteration)

Figure 5: **Effect of Gating.** While a large number of gates (47%) are active at start of training (a, c), they are pruned and only gates corresponding to changing scene content (2%) remain active (b, d).

viewspace gradient differences due to faster convergence of the gates, without loss of quality. By masked training via localized image rendering, we reduce training time by 8 seconds for Immersive and marginally for N3DV. Overall, from the baseline, we obtain significant model size reduction with equivalent or lower training and rendering speed. Attribute quantization framework even improves PSNR compared to the baseline for both datasets. This largely stems from quantizing the scaling attribute leading to a more stable optimization with better reconstruction quality while reducing storage size (Tab. 4).

**Effect of Gating.** As shown in Fig. 5, more than half of the gates are set to be inactive at the start of training with viewspace gradient initialization (Sec. 3.3) and a large portion of the image is active. However, post-training, most gates become inactive while the remaining active gates successfully focus on the dynamic scene content, *e.g.*, the person's hands or the dog's face. This validates that our gating mechanism effectively separates static and dynamic scene content.

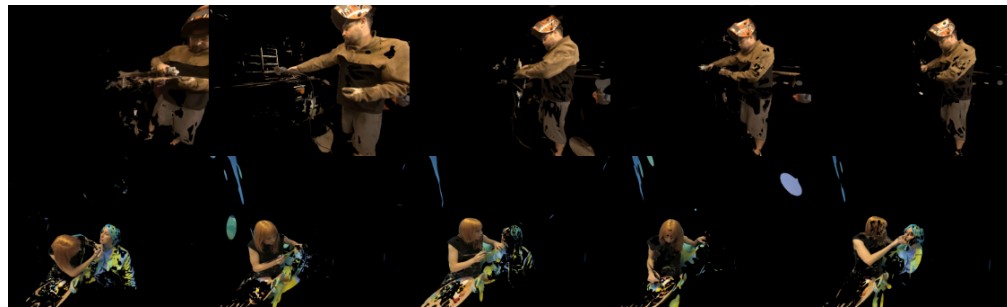

Figure 6: **Adaptive Image Mask Visualization.** We separate out the dynamic scene content at different time-steps of the video through our viewspace gradient difference approach in Sec. 3.3.

**Effect of Adaptive Image Mask.** We visualize the masks obtained by our viewspace gradient difference module in Sec. 3.3. Results on 2 scenes in the Immersive dataset are shown in Fig. 6. For various time instance of the video (columns), we adaptively identify image regions corresponding to the dynamic scene content. We can therefore perform local image rendering and backpropagation for faster training skipping computation for the static parts of the scene such as the background.

## 5 Conclusion

We proposed QUEEN, a framework to model 3D dynamic scenes for online FVV using 3D-GS. We utilized an attribute residual framework, which freely updates all parameters leading to better modeling of complex scenes. We show that the residuals can be successfully compressed via our learned quantization-sparsity mechanism, which adapts to the dynamic scene content to achieve very small model sizes, improved training and rendering speeds, and improved visual quality. In future work, we aim to extend QUEEN for sparse view reconstruction or sequences with long duration.

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

# Appendix

We provide supplementary results (Appendix A), additional implementation details (Appendix B), discussion on limitations and future work (Appendix C) and the broader impact (Appendix D) of our approach. We recommend the reader to watch the supplementary video hosted on our project website: `https://research.nvidia.com/labs/amri/projects/queen` for a visual comparison of the results of the various methods as well as more details of this project.

## A Supplementary Results

### A.1 Quantization vs. Gating

We evaluate the effect of the gating framework in comparison to the quantization framework for the position residuals. We perform gating or quantization on the position residuals while quantizing all other attributes. Results, averaged on the N3DV dataset, are shown in Figure 7. We vary training epochs per frame from 6 to 15 for gating and 6 to 25 for quantization to obtain trade-off curves. Increased training epochs result in higher reconstruction quality (PSNR) but longer training time or a larger model size. The left figure shows the PSNR versus size tradeoff while the right figure shows PSNR versus training time tradeoff. We see that, in both cases, the gating framework produces much better tradeoff curves than quantization, with PSNR values more than 0.2dB higher at similar sizes. When increasing the number of training iterations, quantization still improves in quality albeit at a slower rate requiring more training iterations for convergence. This demonstrates that position attributes are more sensitive to quantization errors and require full precision. It justifies our choice of learning to sparsify them as opposed to quantizing them. However, this does not translate to the other geometric attributes, scaling and rotation, where quantization is sufficient in compressing the attributes. This is seen in the results in Table 5 on the Exhibit scene from the Immersive dataset. Quantizing both rotation and scaling results in the lowest storage memory per frame at a similar PSNR and slightly higher training time.

Table 5: Gating versus Quantization of Rotation and Scale Attributes

| Rotation | Scaling | PSNR | Storage Mem (MB) | Training Time (s) |
|---|---|---|---|---|
| Quantization | Quantization | 29.15 | 2.47 | 21.61 |
| Gating | Quantization | 29.26 | 3.62 | 20.44 |
| Quantization | Gating | 28.92 | 3.64 | 19.89 |

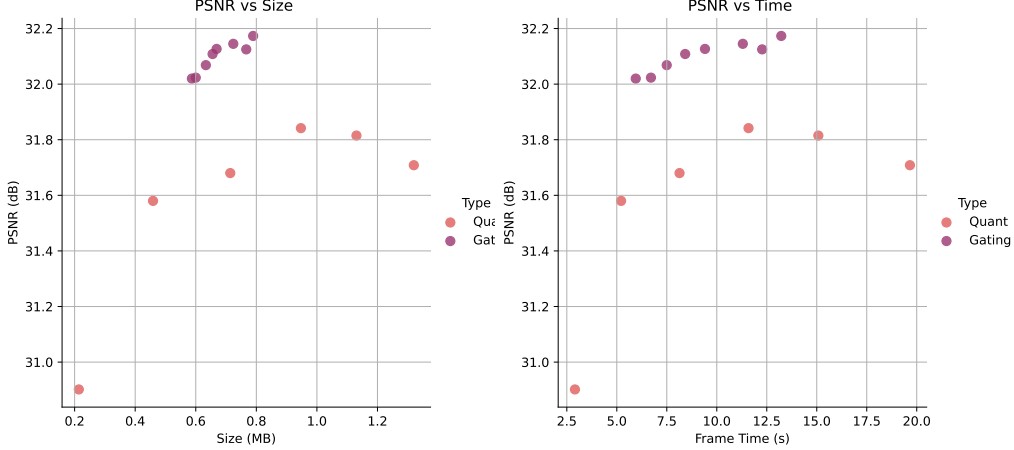

Figure 7: **Effect of Gating vs Quantization of Position Residuals.** Gating leads to much better PSNR versus size or PSNR versus training time tradeoff curves due to higher precision residuals.

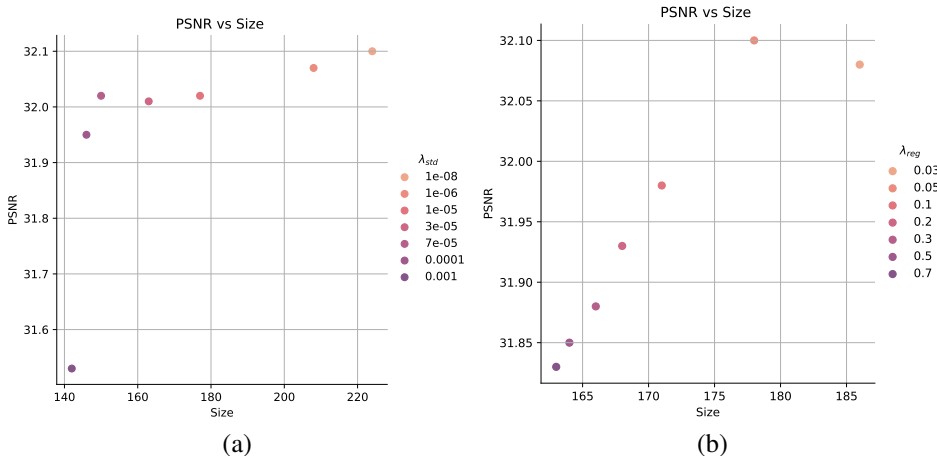

(a)                                        (b)

Figure 8: **PSNR-Size Tradeoffs.** We evaluate the effect of varying $\lambda_{\text{std}}$ and $\lambda_{\text{reg}}$ for (a) the quantized latents and (b) the sparse position residuals. The storage size is measured in KB for the full video duration of 300 frames.

Table 6: **Effect of Latent Dimensions.** Dim is latent dimension and Size is total size of the 300-frame video in MB. *Metrics for Color SH are evaluated on three N3DV scenes.

| Color SH* | | | Color Base | | | Rotation | | | Scaling | | |
|---|---|---|---|---|---|---|---|---|---|---|---|
| Dim | PSNR | Size | Dim | PSNR | Size | Dim | PSNR | Size | Dim | PSNR | Size |
| 1 | 33.69 | 155 | 2 | 32.02 | 211 | 2 | 32.00 | 182 | 2 | 32.13 | 215 |
| 2 | 33.73 | 157 | 4 | 32.04 | 211 | 4 | 31.98 | 183 | 4 | 32.06 | 219 |
| 4 | 33.78 | 156 | 8 | 32.04 | 213 | 6 | 32.01 | 182 | 8 | 32.11 | 224 |
| 8 | 33.71 | 156 | 12 | 32.06 | 215 | 8 | 32.00 | 182 | 12 | 32.07 | 226 |

## A.2    Accuracy-memory Tradeoff

We consider the effect of varying different loss coefficients to trade off between accuracy and memory. In Figure 7 we explore the tradeoff between PSNR and size by varying the number of training iterations. We can also control the amount of sparsity in the scene by varying the $\lambda_{\text{reg}}$ loss coefficient. As visualized in Fig. 8 (b), we find that increasing $\lambda_{\text{reg}}$ leads to higher sparsity or lower memory but also lower reconstruction quality.

We further experiment with an additional regularization loss to reduce the entropy of the latents. We observe that lower entropy corresponds to lower memory, but also lower reconstruction quality. While learnable probability models can successfully reduce entropy, as shown by [25], these models have higher time and memory costs during training. We instead observe that the probability distribution of the various attribute residuals at each time-step is unimodal and is close to a Laplacian or Gaussian distribution. As a unimodal distribution has entropy proportional to the variance [11], we enforce a loss on the standard deviation of the latents with a tradeoff parameter $\lambda_{\text{std}}$ controlling the effect of this regularization loss. Fig. 8(a) shows results on the N3DV dataset by varying $\lambda_{\text{std}}$. We observe that increasing $\lambda_{\text{std}}$ reduces the entropy costs, leading to lower memory costs, but lower reconstruction quality, and vice versa.

## A.3    Effect of Quantization Latent Dimension

We provide additional analysis on the effect of latent dimension for the various attributes in Table 6. In general, latent dimension does not have a significant effect on reconstruction quality or model size. Increasing the latent dimension can lead to lower per-dimension entropy due to our learnable quantization framework and hence still maintains the overall total size for the latent. We find that varying the total number of iterations (Appendix A.1) or the entropy loss/variance coefficient (Appendix A.2) are more effective knobs for trading off between quality-memory or quality-time.

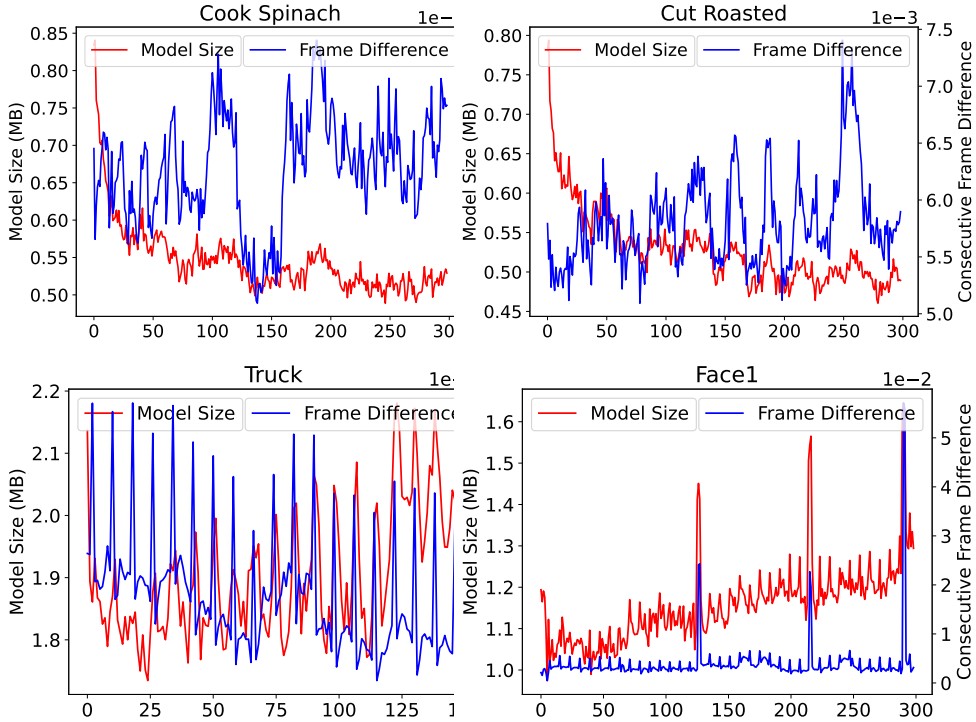

Figure 9: **Adaptive Sizes.** Our quantization-sparsity framework automatically allocates more memory for frames with larger scene changes (higher frame difference). Notice that the frame difference spikes in the Immersive scenes (bottom row) correlate with frame-wise model size, which increases to allow for modeling of more temporal variations.

## A.4 Framewise PSNR and Size

A key advantage of our quantization-sparsity framework is its adaptability to scene content. We visualize the per frame sizes for 2 scenes each from N3DV and Immersive along with the visual frame difference between consecutive frames in Figure 9. We see that our approach allocates variable model sizes for each frame unlike 3DGStream [69], which uses a fixed-sized InstantNGP structure. Additionally, higher frame differences result in larger model sizes and vice versa, as seen in the top row. This shows that our method is capable of allocating more bits to frames with large scene changes. This is especially evident from the spikes in Immersive's scenes in the bottom row, which correlate with the model size.

Next, we show the stability of our approach at recovering from large scene variations corresponding to the frame difference spikes as mentioned above. We visualize the reconstructed test-view PSNR for each frame, for 2 scenes each from N3DV and Immersive, along with the frame difference between consecutive frames in Figure 10. A large L1 error such as around frames 175 (top left), frames 225 (top right), frames 75 (bottom left) or frames 90 and 290 (bottom right) does lead to drops in PSNR. However, our PSNR recovers in subsequent frames showing the stability of our framework with large scene variations present.

## A.5 Effect of Improved Point Cloud Initialization.

Consistent geometry for the 3D scene in the first frame is important to learn accurate residuals for the attributes of the subsequent frames. The COLMAP-generated point cloud initialization can be incomplete for regions that are textureless or are not sufficiently captured in multiple cameras. This is visualized in the top row of Figure 11. The boundaries of the scene consist of limited training view cameras as shown by the white box leading to sparse or no points in these regions by COLMAP initialization. The densification stage in 3DGS is unable to recover from this producing erroneous rendered depth or geometry and also leads to low quality image reconstruction.

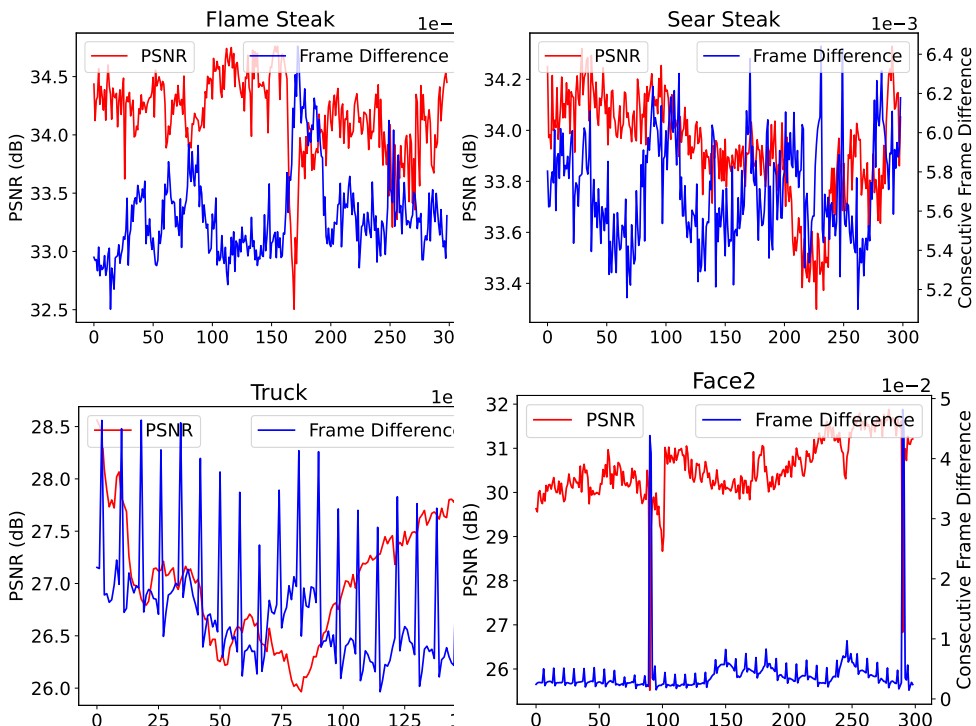

Figure 10: **Per-frame Quality Evaluation.** Our approach results in higher PSNR for large scene changes corresponding to higher consecutive frame difference such as around frame 175 (top right) or the spikes in the bottom right scene.

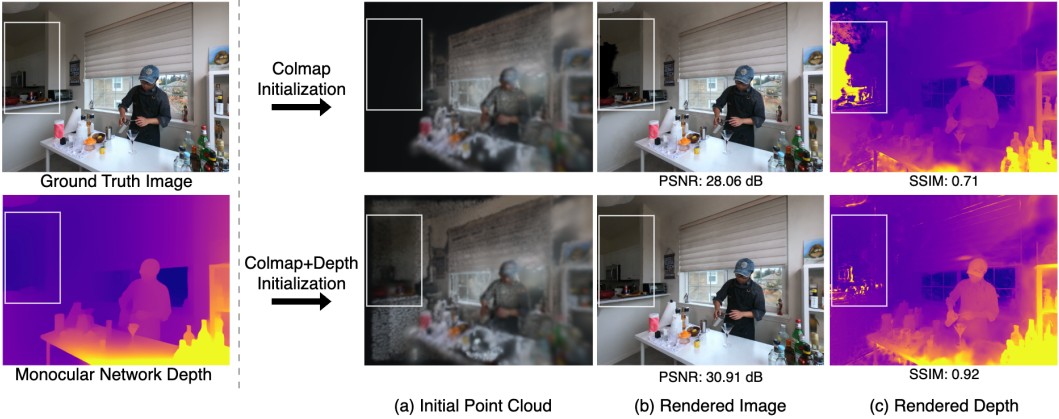

Figure 11: **Effect of Depth Initialization.** Top row: (a) COLMAP produces sparse or no points for regions of the scene with limited texture, producing (b) erroneous image rendering and (c) incorrect geometry or depth. Bottom row: initializing with depth maps predicted by an off-the-shelf monocular depth network produces better reconstruction and consistent scene geometry.

Therefore, we propose to use an off-the-shelf monocular depth estimation network [64] to predict a more complete initial point cloud. However, due to the scale-shift ambiguity of monocular depth estimation, we align the predicted monocular depth with the true scene depth from existing COLMAP points. To do so, we estimate the 2D pixel locations $\mathbf{p}'_i$ of each COLMAP point $i$ by projecting points from 3D world space to 2D screenspace.

$$\mathbf{p}'_i = \Pi(\mathbf{p}_i; \mathbf{K}, \mathbf{W}), \tag{12}$$

where $\Pi(.)$ denotes the perspective transform described in Eq. 1 of the paper. We then query the pixel locations $\mathbf{p}'_i$ in the monocular depth image to obtain the corresponding depth values $\hat{z}_i$. This is

Table 7: **Effect of Depth Initialization on N3DV and Immersive datasets.**

| Dataset | Initialization | PSNR (Test) (dB)↑ | PSNR (Train) (dB)↑ | Num. Points (in Millions) ↓ | Training Time (s) ↓ |
|---------|----------------|-------------------|--------------------|-----------------------------|---------------------|
| N3DV | COLMAP | 32.02 | 30.06 | 3.24 | **13.24** |
| | +Depth | **32.03** | **31.42** | **3.14** | 13.40 |
| Immersive | COLMAP | 28.52 | 32.38 | **3.10** | **18.62** |
| | +Depth | **29.10** | **32.65** | 3.21 | 18.91 |

aligned with the GT depth value from the COLMAP 3D points $z_i$ by a least-squares optimization to obtain the scale and shift parameters $\alpha, \beta$. We then obtain the aligned dense depth map $\alpha \hat{z}_i + \beta$.

To identify regions with empty COLMAP initializations, we iterate over training views and render the corresponding image along with an alpha mask calculating the accumulated transmittance at each pixel location. Mask values below a threshold $t_z$ (0.10 for N3DV and 0.03 for Immersive) are identified to obtain the corresponding pixel locations containing few COLMAP points. We then use the aligned depth values corresponding to these pixel locations to re-project back into the world space.

As seen in the bottom row of Figure 11, the depth map initialization produces more dense points in empty regions. These points maintain consistent depth with existing COLMAP points. Such an initialization results in improved image reconstruction quality with ~3dB improvement in PSNR for the corresponding view while also producing better consistent depth or geometry. As we utilize the network depth at initialization only, we do not require high-quality depth networks and a coarse depth is sufficient for sampling new points. The training can then learn to move the Gaussians to produce finer scene depth. This also results in minimal increases in training time as it's a one-time operation at the initial time-step and produces a small number of additional points in empty regions.

We show quantitative results for 2 configurations with and without depth map initialization for the datasets of N3DV and Immersive in Tab. 7. We show PSNR for the central test view as well as for the first train view for all scenes. We also show the number of points at the end of training the first frame with and without initialization along with the average training time per frame for the full scene. The depth initialization does not significantly affect the test view PSNR on N3DV as it corresponds to the central view with a large overlap with many training views. In contrast, the PSNR for the training view (also in Fig. 11) improves considerably (+1.3dB) with the depth initialization highlighting the efficacy of the approach in improving geometry for regions with sparse camera views. Additionally, there is almost no overhead cost as we obtain similar number of Gaussian points at the end of training the first frame. Training time for the full video is also only marginally higher. On Immersive, we observe a 0.5 dB improvement in PSNR for the test view while the train view PSNR shows minor improvements. Again, the number of Gaussians and training time is not significantly affected by the initialization with additional depth based points.

## A.6 Additional Baseline Comparisons

We show comparisons to additional offline FVV baselines on the N3DV dataset in Table 8 for the sake of completeness. This is a superset of the Tab. 2 in the main paper. A majority of the works compute SSIM using the scikit-image implementation, which tends to produce higher values different from our SSIM implementation similar to MipNeRF [2]. Since the two are not comparable, we exclude implementation numbers computed with scikit-image. We also only show results for LPIPS on VGG for the methods that use it. This is consistent with how we compute LPIPS.

Table 8: **Quantitative Comparisons on the N3DV [41] and Immersive [4] Datasets.** We compare QUEEN against state-of-the-art online and include offline FVV methods for completeness. 3DGStream* refers to our re-implementation on the same NVIDIA A100 GPU as used by QUEEN for fairness. $^{\dagger}$ is evaluated on the *flame salmon* scene only. Bold and underlined numbers indicate the **best** and the second best results, respectively, within each category.

| Type | Method | PSNR (dB) ↑ | SSIM ↑ | LPIPS ↓ | Storage (MB) ↓ | Training (sec) ↓ | Rendering (FPS) ↑ |
|------|--------|-------------|--------|---------|----------------|------------------|-------------------|
| Offline | DyNeRF [41] | $29.58^{\dagger}$ | $0.961^{\dagger}$ | $0.083^{\dagger}$ | **0.1** | 15600 | 0.02 |
| | NeRFPlayer [68] | 30.69 | 0.932 | 0.209 | 17.1 | 72 | 0.05 |
| | HyperReel [1] | 31.10 | 0.928 | - | 1.2 | 104 | 2.00 |
| | HexPlanes [5] | 31.70 | - | - | 0.8 | 144 | 0.21 |
| | K-Planes [19] | 31.63 | - | - | 1.0 | 48 | 0.15 |
| | MixVoxels [76] | 31.34 | - | - | 1.7 | 16 | 38 |
| | 4DGS [88] | 32.01 | - | - | - | - | 114 |
| | HexPlanes-4DGS [83] | 31.15 | - | - | 0.3 | **6** | 30 |
| | SpaceTime [42] | **32.05** | **0.948** | - | 0.67 | 20 | **140** |
| Online | StreamRF [37] | 30.68 | - | - | 31.4 | 15 | 8.3 |
| | TeTriRF [84] | 30.43 | 0.906 | 0.248 | **0.06** | 39 | 4 |
| | 3DGStream [69] | 31.67 | - | - | 7.83 | 12 | 215 |
| | 3DGStream* [69] | 31.58 | 0.941 | 0.140 | 7.80 | 8.5 | 261 |
| | QUEEN-s | 31.89 | 0.945 | 0.139 | 0.68 | **4.65** | **345** |
| | QUEEN-m | 32.03 | 0.946 | 0.137 | 0.69 | 5.96 | 321 |
| | QUEEN-l | **32.19** | **0.946** | **0.136** | 0.75 | 7.9 | 248 |

## A.7 Per-scene Results

In addition to the average quantitative results over the full datasets of N3DV and Immersive in Tab. 8, we show results for each scene in both the N3DV and Immersive datasets of various frame-wise metrics, including, PSNR, SSIM, LPIPS, size, training time and rendering time (FPS) in Tables 9 and 10.

## A.8 Perceptual quality: User study

In addition to the extensive quantitative analysis in the paper as well as supplementary, we conduct an A/B user study to measure the perceptual quality of our video reconstructions. For each vote, we show a pair of randomly chosen rendering results (from a test view that is not used in training) by our method and one of the baseline methods (3DGStream [69] and TeTriRF [84]). We also show the ground truth video as a reference for the participants to make the decision. We ask the participant to choose the method that more faithfully matches the reference video. In total, we collected 285 responses from 15 participants within the timeline of the rebuttal. For the N3DV dataset, 76.67% of users preferred our method over 3DGStream and 96.67% preferred our method over TeTriRF. On the Google Immersive dataset, 97.14% of users preferred the reconstructions from our approach over that of 3DGStream. This showed that the participants strongly prefer our results in comparison to the baseline methods for both datasets.

## B  Implementation Details

**Training.** Our implementation of QUEEN builds on that of [29]. We train the Gaussians for 500 and 350 epochs for first time-step, and for 10 and 15 epochs for the subsequent time-steps, on N3DV and Immersive, respectively, with each epoch consisting of all training views. We set the SH degree to 2 for N3DV and 3 for Immersive. We set the score vector threshold $t_{\mathbf{d}} = 0.001$ for all experiments. We additionally dilate the image mask by a $48 \times 48$ kernel to include neighboring image regions while rendering as larger Gaussians can depend on multiple pixel locations. We perform masked training for 30% of the iterations for N3DV and 65% for Immersive. We perform masked training for only a fraction of iterations as updating Gaussians rendered only at masked locations can alter the rendered pixels at unmasked location as well. We thus allow fine-tuning on the full image to account for any changes in the unmasked image regions.

Table 9: **Per-scene Metrics for the N3DV Datasets**

| Scene | PSNR (dB) ↑ | SSIM ↑ | LPIPS ↓ | Storage (MB) ↓ | Training (sec) ↓ | Rendering (FPS) ↑ |
|---|---|---|---|---|---|---|
| Coffee Martini | 28.38 | 0.915 | 0.155 | 1.17 | 7.48 | 213 |
| Cook Spinach | 33.4 | 0.956 | 0.134 | 0.59 | 8.03 | 254 |
| Cut Beef | 34.01 | 0.959 | 0.132 | 0.57 | 7.59 | 291 |
| Sear Steak | 33.93 | 0.962 | 0.125 | 0.56 | 9.31 | 257 |
| Flame Steak | 34.17 | 0.962 | 0.126 | 0.59 | 7.97 | 266 |
| Flame Salmon | 29.25 | 0.923 | 0.145 | 1.00 | 7.00 | 207 |
| Average | 32.19 | 0.946 | 0.136 | 0.75 | 7.90 | 248 |

Table 10: **Per-scene Metrics for the Immersive Datasets**

| Scene | PSNR (dB) ↑ | SSIM ↑ | LPIPS ↓ | Storage (MB) ↓ | Training (sec) ↓ | Rendering (FPS) ↑ |
|---|---|---|---|---|---|---|
| Welder | 27.03 | 0.884 | 0.215 | 2.18 | 18.23 | 146 |
| Flames | 30.52 | 0.925 | 0.157 | 1.26 | 29.54 | 161 |
| Truck | 27.03 | 0.905 | 0.195 | 2.03 | 17.12 | 177 |
| Exhibit | 28.03 | 0.903 | 0.193 | 2.11 | 17.81 | 170 |
| Face Paint 1 | 31.90 | 0.950 | 0.240 | 1.19 | 17.48 | 245 |
| Face Paint 2 | 30.55 | 0.937 | 0.207 | 2.39 | 17.49 | 217 |
| Cave | 29.49 | 0.900 | 0.250 | 1.50 | 20.46 | 162 |
| Average | 29.22 | 0.915 | 0.208 | 1.79 | 19.73 | 183 |

**Sparse Gating.** We learn the parameters $\alpha$ with the Adam optimizer [30]. The position residual learning rate is set to 0.00016 for N3DV and 0.0005 for Immersive. Other hyperparameters are provided in Table 11.

**Quantization.** For both datasets, we set the learning rate of the decoder parameters to be 0.001 for the color and rotation attributes and 0.0001 for opacity and scaling, optimized with the Adam optimizer. Other hyperparameters are provided in Table 12.

**Densification.** For N3DV, we perform densification for subsequent frames from epoch 6 until $80\%$ of the epochs with an interval of 2 epochs and a gradient threshold of 0.00125. For Immersive, we perform densification on the 8th epoch with a gradient threshold of 0.00125.

Table 11: **Gating Hyperparameters**

| Dataset | LR | $\lambda_{\text{reg}}$ | $\gamma_0$ | $\gamma_1$ | $\tau$ |
|---|---|---|---|---|---|
| N3DV | 0.1 | 0.01 | -0.5 | 1.01 | 0.3 |
| Immersive | 0.1 | 0.01 | -0.1 | 1.1 | 0.5 |

**First-frame Quantization.** Uncompressed Gaussians have large memory costs even for the first frame. However, quantizing all attributes results in quality degradations [24]. We therefore apply learnable quantization only on the first frame's high-frequency spherical harmonic coefficients (excluding the DC component) using the hyperparameters mentioned above. For example, on N3DV, we reduce the first frame's size from 47 to 17 MB with no quality degradation.

**Storage.** Post-training, we convert the learned parameters to the final compressed form. For the quantized residuals of rotation, scale, opacity and appearance, we convert the continuous latents $\mathbf{L}_c$ to the integer form and then apply entropy encoding and store this further-compressed representation $\mathbf{E}_c$ as well as the decoder $\mathbf{D}_c$ for all four categories $c \in \{\text{r, s, o, } \mathbf{h}\}$. Our entropy coding approach flattens our integer latent matrix for each attribute before encoding. For example, for L-dimensional latent attributes for N gaussians, we flatten the matrix to obtain a vector with L*N elements. This integer vector is then encoded using standard entropy coding approaches such as arithmetic coding.

Table 12: **Quantization Hyperparameters**

| Dataset | Rotation | | Scaling | | Opacity | | Color Base | | Color Freq | |
|---------|----------|----|---------|----|---------|----|------------|----|------------|----|
| | Latent Dim | Latent LR | Latent Dim | Latent LR | Latent Dim | Latent LR | Latent Dim | Latent LR | Latent Dim | Latent LR |
| N3DV | 6 | 0.025 | 8 | 0.01 | 3 | 0.05 | 8 | 0.0125 | 4 | 0.000625 |
| Immersive | 6 | 0.015 | 8 | 0.007 | 3 | 0.05 | 8 | 0.0125 | 12 | 0.000375 |

The number of bits for storing each attribute residual matrix is therefore dependent on the scene content as it relies on the amount of motion. This number can be fractional, on average, which is the standard for the entropy coding algorithm of arithmetic coding or Huffman coding [33]. For example, for the Sear Steak scene in the N3DV dataset, on average, we require 0.68 bits for all the quantized attributes (corresponding to 0.5MB/frame). This depends on the entropy of the latents itself, which varies with changing scene motion (Figure 9).

For the sparse gates, we store the positional residuals as a sparse matrix with the indices from binarized gate variables $\mathbf{I} = \{i | g_i \neq 0\}$, and the full-precision residual vectors only if its corresponding gate is on, $\mathbf{E}_p = \{\mathbf{l}_{p_i} | i \in I\}$. Both operations add a negligible computation overhead. This corresponds to the coordinate format (COO) for storing sparse matrices where the non-zero values are stored in FP-32 precision along with their integer index locations.

## B.1 Sensitivity Analysis on Hyperparameter

Table 13: **Sensitivity to different hyperparameter configurations on the N3DV dataset.**

| Method | PSNR | Size (MB) |
|--------|------|-----------|
| Ours (N3DV hyperparam.) | 32.14 | 0.60 |
| Ours (Immersive hyperparam.) | 32.06 | 1.49 |
| 3DGStream [69] | 31.58 | 7.80 |

We set different hyperparameters for the two datasets in in Tables 11 and 12 to account for the widely varying amount of scene motion between N3DV and Immersive datasets. The Immersive dataset contains larger and more complex scene motions (e.g., a person entering and leaving the scene) while N3DV contains relatively minor motions. We found that a higher learning rate for the position residuals allows Gaussians to adapt to the highly dynamic scenes. The gating hyperparameters in Table 11 for N3DV are set to utilize this prior information about the dataset where the stretch hyperparameters $\gamma_0$ and $\gamma_1$ are set closer to 0 to enforce more sparsity in the position residuals. Additionally, the Immersive dataset itself consists of a wide variety of indoor/outdoor scenes at varying scales/scene motion/illumination. We use the same set of hyperparameters for each scene achieving good reconstruction quality for every scene (Table 10) showing its generalization capability.

Tables 11 and 12 list out the different hyperparameters for quantizing and sparsifying residuals for both N3DV and Immersive datasets. To test the sensitivity of reconstruction quality to hyperparameters, we train our method on the N3DV dataset with two sets of hyperparameters. The first configuration uses the stated hyperparameters for N3DV from Tables 11 and 12, while the second utilizes the hyperparameters corresponding to Immersive while also matching the learning rate for the position residuals (0.0005). We show results on N3DV datasets in Table 13. We see that the Immersive dataset's hyperparameter configuration still achieves similar PSNR as the original hyperparameters for N3DV. While the model size is higher (1.49 MB) with the Immersive configuration compared to the original configuration (0.60 MB), it is still much lower than the prior state-of-the-art 3DGStream (7.8 MB) while maintaining higher reconstruction quality in terms of PSNR.

## B.2 Evaluation

**Datasets. (1) Neural 3D Video (N3DV) Datasets** [41] consist of six indoor scenes with forward-facing multiview videos with up to 20 cameras at $2704 \times 2028$ resolution. Similar to prior work, we downsample videos by a factor of 2 for training and testing, holding out the central view for testing. Each video consists of 300 frames at 30 FPS. **(2) Immersive Video Datasets** [4] consist of light field videos of indoor and outdoor scenes captured using a 46-camera rig with fisheye lenses. Following

prior work, we downsample videos by a factor of 2 to obtain a resolution of $1280 \times 960$. We evaluate on 7 scenes (Welder, Flames, Truck, Exhibit, Face Paint 1, Face Paint 2 and Cave) with the central view held out for testing. We extract the first 300 frames for all scenes except for Truck, which consists of 150 frames. We undistort the fisheye views into perspective views using the distortion parameters. We train and evaluate on the perspective views with pinhole camera parameters.

**Baselines. 3DGStream [69]:** We use the official codebase[4] from 3DGStream [69]. We use the same default configuration for N3DV as provided by the authors. For Immersive, we reduce the gradient threshold for densification to 0.0075 to allow for more Gaussians while increasing the training iterations for Stage 1 and 2 to be 450 and 250 iterations, respectively. **TeTriRF [84]:** We use the official codebase[5] from TeTriRF [84] for all experiments on N3DV.

**Measuring FPS.** We compute FPS for all frames of the video and report the median value in our experiments including the time taken to decode the residuals. Note that the decoding is a one-time operation per step and rendering for a preceding frame can be performed while learning the residuals of a subsequent ones to achieve even higher speeds.

## C  Limitations and Future Work

For efficiency and on-the-fly training, we encode sequences by learning inter-frame residuals. However, for FVVs of long duration or drastic scene update, per-frame training will face challenges in reconstruction capability. Unlike offline reconstruction, per-frame training does not have access to future-frame information. This setup limits the capability to effectively reason about large scene changes (e.g., topological changes and highly varying appearance) [56]. If an object suddenly appears or disappears, it is more difficult to (de-)allocate and update scene parameters to capture such changes. In the context of Gaussian splatting, it is challenging to schedule densification and pruning of the Gaussians. Future work could address this by designing an efficient keyframing technique for identifying large changes in the scene and allocate longer training times accordingly.

Furthermore, most current FVV encoding paradigms rely on the input multi-view videos for reconstruction. Exploring a general prior of the dynamic scenes, e.g., generative video models, is a promising direction for reducing the dependence on coherent multi-view input, as the video prior could regularize the reconstructed FVV to capture reasonable scene dynamics even if some input views or frames are missing. Moreover, extending our approach to a single or sparse view scenario is a challenging yet important problem for further democratizing streamable FVV. We will leave these directions for future work.

## D  Broader Impacts

We consider our work as a neutral technology. This proposed method reconstructs free-viewpoint videos from user-provided video inputs. As we highlighted in the introduction, this technology can improve many aspect of people's lives, such as through healthcare (tele-operation) and communications (3D video conferencing). There is indeed a possibility that this work can be misused. Since our reconstruction completely relies on the video inputs, the most likely cases of misuse are those where the input video (provided by the users) have negative impacts.

---

[4]https://github.com/SJoJoK/3DGStream
[5]https://github.com/wuminye/TeTriRF

