# OpenReview forum: "QUEEN: QUantized Efficient ENcoding of Dynamic Gaussians for Streaming Free-viewpoint Videos"
_NeurIPS.cc/2024/Conference — NeurIPS 2024 poster_

### Official Review · Reviewer_9x86 · 2024-06-30

**Soundness:** 3
**Presentation:** 3
**Contribution:** 3
**Rating:** 7
**Confidence:** 4

**Summary:**

This paper introduces a novel technique, Quantized Efficient Encoding (QUEEN), to achieve streamable free-viewpoint videos. Unlike methods that directly optimize per-frame 3D-GS given multi-view videos, QUEEN learns the residuals of Gaussian attributes between continuous frames by decoding learnable latent embeddings for each 3D Gaussian. Additionally, a quantized decoder technique has been adopted to significantly reduce the size of latent embeddings (except for the embedding of 3D positions), with only a slight or no decrease in rendering performance. As the authors observe that position residuals are sensitive to compression, they introduce a differentiable L0 sparsity penalty (hard concrete gate with learnable parameters) instead of quantizing the position embeddings. Finally, to improve training efficiency and reduce storage requirements, the values of the gates for each frame are initialized using the 2D viewspace Gaussian gradient difference between neighboring frames. Extensive experiments have shown that QUEEN achieves the best training time, model size, and rendering performance compared to state-of-the-art online free-viewpoint video methods.

**Strengths:**

The paper introduces a novel framework, Quantized and Efficient Encoding (QUEEN), which leverages 3D Gaussian Splatting (3D-GS) for streamable free-viewpoint video (FVV). The combination of 3D Gaussian and the quantized optimization is both novel and original. A comprehensive performance analysis of quantization and the sparsity loss has been conducted to validate this technique's design choices and effectiveness. This work provides several key insights for compressing a sequence of 3D Gaussians, achieving the best trade-off between quality, storage, and speed through quantization and sparsification. Compared to state-of-the-art offline and online methods, QUEEN significantly outperforms in training speed, storage efficiency, and rendering speed, while its rendering quality remains comparable to offline methods.

**Weaknesses:**

1. Section 3.2 is difficult to follow due to the absence of preliminary information. There is a lack of detail regarding the quantization process. Specifically, the exact implementation of the quantization that results in more than a 10x size reduction is not provided. A detailed ablation study of the quantization type and the length of latent embeddings would be beneficial. Additionally, since the authors do not claim they will release the code for QUEEN, it is challenging for other researchers to evaluate and validate QUEEN's performance.
2. The ablation results presented in Table 2 are confusing and lack crucial analysis. For instance, it is unclear why the baseline achieves the highest rendering FPS without attribute quantization on the Immersive Dataset, while the same quantization improves the rendering FPS on the N3DV dataset, which appears contradictory. Furthermore, there is no clear explanation for why quantization improves rendering PSNR but results in slower training speeds. Additional analysis is needed to clarify these observations.

**Questions:**

1. Please clarify and add more details about the residual quantization in Section 3.2 to make it easier to follow. It would be beneficial to include an ablation study comparing different quantization methods. Additionally, highlighting any specific technical contributions of this work compared to other quantization approaches would improve the readability and comprehension of this section, or this is just a novel application of quantization technique?
2. Please provide a more detailed analysis of the ablation study results to better understand the key design choices in QUEEN’s framework. This additional analysis will help elucidate how these design choices contribute to the performance and effectiveness of the proposed method.

**Limitations:**

The authors have addressed the limitations of QUEEN as well as the societal impacts in their supplementary material.

---

> ### Author Rebuttal · Authors · 2024-08-07
>
> Thank you for your thoughtful feedback. Please refer to our shared rebuttal (texts and PDF) for additional discussion and results. We address the specific questions below.
>
> ***
>
> ## Q: Details and ablation studies regarding the quantization process.
> Thank you for the suggestion. We will add a detailed explanation on the quantization process in the revision as preliminary.
>
> - (a) We provide a more detailed explanation on entropy coding (under “Q: No information on how the entropy coding works” for Reviewer dVjq). We answer questions about the quantization approach (under “Q: Experimental settings” for Reviewer jnXV). Please refer to corresponding sections for the details. We will clarify in the revision.
>
> - (b) We provide further explanation and ablations analyzing the accuracy-memory tradeoffs. In the PDF, we show results varying loss coefficients for the quantization and sparsity modules in Fig. R1. We also ablate the effect of varying latent dimension in Tab. R1. We discuss these results in the shared responses (under “Q: Accuracy-memory trade-off and ablation study on quantization setup”).
>
> ***
>
> ## Q: Highlighting specific technical contributions.
>
> We agree that the ideas behind the individual modules for quantization and sparsity have been used for other applications such as static scene reconstruction and deep network compression. Our contribution is that we propose an approach for modeling dynamic Gaussians as residuals combined with the optimization of their storage size via our quantization-sparsity framework. This has not been explored previously. We will clarify our contribution in the revision.
>
> ***
>
> ## Q: Code release.
> Yes, we will release the code for research purposes upon acceptance.
>
> ***
>
> ## Q: The ablation results presented in Tab. 2 are confusing and lack crucial analysis.
> Thank you for pointing it out. We will address this further in the paper. Please refer to the shared responses for all reviewers (under "Q: Evaluation analysis and confusing Tab. 2").

---

> > ### Comment · Reviewer_9x86 · 2024-08-09
> >
> > Thank you for the detailed reply. Most of my concerns have been addressed. However, I remain unclear about the significant decrease in FPS on the Immersive dataset, as shown in Table 2. Could the authors please clarify why a more aggressive Gaussian densification has a different impact on the baseline compared to the baseline with attribute quantization? I would appreciate further elaboration on this point.

---

> > > ### Comment · Area_Chair_6sVo · 2024-08-10
> > >
> > > Thank you for the discussion.
> > > Any other thoughts from other reviewers?

---

> ### Author Response · Authors · 2024-08-14
>
> Thank you for initiating discussions, reviewers and AC. Thank you for the insightful comment. Rendering speed for the Gaussian splatting depends on a wide variety of factors involving total number of Gaussians, their characteristics in terms of position, rotation, scale, and opacity as the rasterization for each tile varies depending on these attributes. Hardware configurations (I/O, OS scheduling, GPU clock fluctuation) can also cause a difference. For a similar number of Gaussians, it is possible for FPS to vary drastically with different attribute distributions which is indeed what we observe for N3DV (Table E-1(b)) based on the different scaling attribute distribution (Figure R2).
>
> **Table E-1**: Further analysis on rendering speed. We provide more comparison between the baseline and QUEEN (+Attribute Quantization) on (a) one sequence “flame” from the google immersive datasets and (b) the N3DV datasets. All time measures are in the unit of milliseconds. Storage size in MegaBytes (MB).
>
> (a) Sequence “flame” from the google immersive datasets
> | Config                    | PSNR (dB) | Storage Size (MB)| Rendering time (ms) / FPS | Decoding Time (ms) / FPS | Num. Gauss. |
> | ------------------------- | --------- | ----------- | --------------------- | -------------------- | ----------- |
> | Baseline                  | 28.14     | 16434       | **4.13 / 242**    | N/A             | **217K**   |
> | \+ Attribute Quantization | **30.76** | **1030**    | 4.47 / 223.9      | 0.43 / 2293     | 240K       |
> | Relative Increase         | +9.3%     | \-93.7%     | +8.1%             | N/A             | +10.6%     |
>
> (b) N3DV datasets
>
> | Config                    | PSNR (dB) | StorageSize (MB)| Rendering time (ms) / FPS | Decoding Time (ms) / FPS | Num. Gauss. |
> | ------------------------- | --------- | ---------- | --------------- | ------------ | ---------- |
> | Baseline                  | 31.66     | 13308      | 4.67 / 214        | N/A          | **311K**   |
> | \+ Attribute Quantization | **32.04** | **1255**   | **3.51 / 285**    | 0.43 / 2337    | 311K       |
> | Relative Increase         | +1.2%     | \-90.8%    | \-24.8%         | N/A          | +0%        |
>
> Table E-1 shows a more extensive analysis on rendering efficiency and other performance metrics of the baseline and with attribute quantization, for the sequence “flames” in Immersive as well as on the N3DV dataset. On the sequence “flames” of the immersive datasets, we obtain 10.6% more Gaussians (+23k) with the densification process compared to the baseline. This leads to higher rendering time (or lower FPS) (+8.1%) which is increased further due to the additional decoding time (0.43 ms).
>
> Our attribute quantization of the Gaussians during training affects the distribution of the attributes, leading to different cloning and splitting during densification (based on viewspace gradients and Gaussian scale attribute) compared to the baseline. This also varies with different scenes depending on the scene changes (e.g., motion and appearance/disappearance of objects).
>
> We highlight that the main goal of our paper is to reduce the storage and training time of online free-viewpoint videos (FVVs) while maintaining the reconstruction quality and the efficient real-time rendering speeds brought by Gaussian splatting. Although it is interesting to pursue higher rendering efficiency or controllability, those are not our goal. Our approach does not explicitly control the distribution of the Gaussian attributes or the number of Gaussians, which are the main factors behind rendering speeds. We believe this can be an interesting direction for future work to pursue and we will add discussion in the revision.
>
> Furthermore, our method addresses a joint problem of reconstruction and quantization for online FVV, in contrast to a two-step formulation of first reconstruction and then quantization. In the latter case, it is more likely for the system to pay for additional computational cost and some loss of quality. In our case (the former case), a joint formulation shows more potential to achieve a better tradeoff between quality and efficiency, as shown in Tab. D1 as well as recent methods such as [84]. We will add discussion in the revision to better explain our contributions.
>
> We appreciate the discussion and we will include it in the revision for future audiences. Furthermore, we will release the codebase to the community to experiment with various types of scenes and discover new insights into the effect of quantization on rendering speeds.

---

> > ### Comment · Reviewer_9x86 · 2024-08-14
> >
> > Thanks for authors' thorough analysis of the trade-offs between rendering speed, quality, and quantization. Since my concerns have been fully addressed, I would like to raise my rating to a 7.

---

### Official Review · Reviewer_L5E8 · 2024-07-09

**Soundness:** 3
**Presentation:** 3
**Contribution:** 3
**Rating:** 6
**Confidence:** 2

**Summary:**

This paper presents a novel method for free-viewpoint video learning and streaming based on 3D Gaussian splatting (3DGS). The proposed method learns the attribute residuals of the raw Gaussian points in a frame-by-frame fashion, and the learning process is incorporated with both latent quantization and sparsity regularization. In addition, dynamic contents are identified and separated through Gaussian viewspace gradient difference in order to further accelerate computation. Compared to existing works, the proposed method is able to reduce the memory cost of FVV by a large margin, while preserving high-quality representation capability.

**Strengths:**

* This paper is overall a solid submission. It addresses the challenging problem of FVV encoding and compression. The proposed method is able to a high compression rate while ensuring reconstruction quality.

* The authors conduct comprehensive comparisons against state-of-the-art methods. The experiments are convincing.

* The paper is overall well-written and easy to follow.

**Weaknesses:**

* The idea of exploiting temporal redundancy via encoding attribute residuals is not new. Although it has not been explored in the context of 3DGS, similar ideas have already been studied in NeRF-based representation, such as [75].

* The proposed method appears to be highly sensitive to hyperparameter tuning, as indicated in Section B, Tables 9 and 10. The authors find it necessary to adjust hyperparameters for different datasets. For instance, the learning rate for position residual learning in Immersive (0.0005) is approximately twice as high as that in N3DV (0.00016). Other hyperparameters, such as gating and quantization parameters, also exhibit significant differences across these datasets.

**Questions:**

None

**Limitations:**

The authors have discussed the limitation and potential social impact Sec.C and Sec.D of the supplemental document, respecitively.

---

> ### Author Rebuttal · Authors · 2024-08-07
>
> Thank you for your thoughtful feedback. Please refer to our shared rebuttal (texts and PDF) for additional discussion and results. We address the specific questions below.
>
> ***
>
> ## Q: The idea of exploiting temporal redundancy via encoding attribute residuals is not new.
>
> Thank you for confirming the value of our exploration on temporal redundancy with 3DGS. While we agree that the idea of residual training has been explored in some way for NeRFs in [75], it is not trivial to adopt residual modeling on 3DGS due to its explicit nature and lack of overall structure. Furthermore, a key contribution of ours comes from not just modeling dynamic Gaussians as residuals but combining it with the optimization of their storage size via our quantization-sparsity framework (as also mentioned by Reviewer 9x86).
>
> ***
>
> ## Q: The proposed method appears to be highly sensitive to hyperparameter tuning.
>
> **Tab. D3**: Sensitivity to hyperparameters.
> | Method                        |  PSNR    | Size (MB) |
> | :---------------------------- | -------: | --------: |
> | Ours (orig N3DV hyperparam.)  |  32.14   | 0.60 |
> | Ours (Immersive hyperparam.)  |  32.06   | 1.49 |
> | 3DGStream                     |  31.58   |  7.80 |
>
> - (a) To test the sensitivity of reconstruction quality to hyperparameters, we train our method on the N3DV dataset with two sets of hyperparameters. The first configuration uses the original set of hyperparameters while the second utilizes the hyper parameters corresponding to Immersive in Tab. 9 and 10 while also matching the learning rate for the position residuals (0.0005). We show results on N3DV datasets in Tab. D3. We see that the Immersive hyper parameter configuration still achieves similar PSNR as the original set for N3DV. While the model size is higher (1.49 MB) with the Immersive configuration compared to the original configuration (0.60 MB), it is still much lower than the prior state-of-the-art 3DGStream (7.8 MB) while maintaining higher reconstruction quality in terms of PSNR.
>
> - (b) Most of the hyperparameter difference stems from the widely varying amount of scene motion between N3DV and Immersive datasets. The Immersive dataset contains larger and more complex scene motions (e.g., a person entering and leaving the scene) while N3DV contains relatively minor motions. We found that a higher learning rate for the position residuals allows Gaussians to adapt to the highly dynamic scenes.
>
> - (c) The gating hyperparameters in Tab. 9 for N3DV are set to utilize this prior information about the dataset where the stretch hyperparameters $\gamma_0$, $\gamma_1$ are set closer to 0 to enforce more sparsity in the position residuals.
>
> - (d) Additionally, the Immersive dataset itself consists of a wide variety of indoor/outdoor scenes at varying scales/scene motion/illumination. We use the same set of hyperparameters for each scene achieving good reconstruction quality for every scene (Tab. 7) showing its generalization capability.
>
> - (e) Furthermore, we will release the code for the community to experiment.

---

### Official Review · Reviewer_dVjq · 2024-07-10

**Soundness:** 4
**Presentation:** 3
**Contribution:** 3
**Rating:** 5
**Confidence:** 4

**Summary:**

This paper proposes a framework called QUEEN, based on Gaussian splatting, for compact free-viewpoint videos. The data size is reduced to around 0.7MB per frame while achieving fast training speeds. QUEEN encodes the Gaussian attribute changes between consecutive frames. Specifically, it uses latent codes to embed the attribute residuals and employs a learnable gating scheme to update only the Gaussians that exhibit movement. The authors also introduce methods such as gate initialization and point cloud initialization for improved performance. The results are promising.

**Strengths:**

QUEEN updates all Gaussian attributes and achieves better performance than those that do not. The position residual gating is introduced to reduce the computational complexity caused by full updating, which is efficient.

The entire framework is sound. The authors also provide comprehensive experiments to demonstrate its effectiveness and justify the choice of the combination.

**Weaknesses:**

This experiment was conducted on Neural 3D Videos and Immersive Videos datasets, which feature forward-facing scenes with limited viewing angles. When we talk about free-viewpoint videos, we also expect large freedom for the novel viewpoints. However, this paper does not demonstrate such scenarios, such as 360-degree rendering for dynamic objects. The authors claim that QUEEN is intended for streaming FVVs, but the experiments do not evaluate or demonstrate the "streaming" feature.

There is not information that introduces how does the entropy coding work on the quantized integer latents.

**Questions:**

The reviewer suggests that the authors make their claims  more accurate, as mentioned in the weaknesses section.

Some notations are confusing. \mathbf{l}_i denotes the quantized integer latents, while \mathbf{l}_{p_i} is the learnable pre-gated residual. In line 223, \alpha is learnable parameter.  what is the relationship between  \alpha and g_i ?

**Limitations:**

In the limitations section, the authors mention that per-frame training faces challenges in reconstruction capability. It would be helpful if the authors could provide a more detailed explanation.

---

> ### Author Rebuttal · Authors · 2024-08-07
>
> Thank you for your thoughtful feedback. Please refer to our shared rebuttal (texts and PDF) for additional discussion and results. We address the specific questions below.
>
> ***
>
> ## Q: 360-degree rendering.
>
> We evaluate on N3DV and Immersive datasets as they are widely-adopted standard benchmarks for various prior works [6,7,8,10,11,12,16] on free-viewpoint videos.
>
> While these two datasets indeed contain forward-facing scenes, there are limited 360-degree datasets in the research community. Most of the 360-degree datasets focus on human characters or single objects (e.g., D-NeRF, Ava-256), which is a different focus from our goal to capture scene-level dynamics. To the best of our knowledge, N3DV and immersive datasets are among the best benchmarks for real-world scene details and dynamics.
>
> Note that our approach makes no assumptions on a forward-facing setup and can directly be applied to 360-degree scenes. In the revision, we will evaluate our method on the CMU Panoptic studio datasets (http://domedb.perception.cs.cmu.edu/), which covers 360-degree and contains some dynamics beyond a single human character.
>
> ***
>
> ## Q: No evaluation or demonstration of the "streaming" feature.
>
> The streaming feature is inherently built into our method. Due to the per-frame reconstruction and encoding, at each time-step, we only need to transmit the corresponding residuals with no dependence on future frames. Furthermore the high compression rate and high training/rendering efficiency are essential features for streaming.
>
> ***
>
> ## Q: No information on how does the entropy coding work.
>
> Our entropy coding approach flattens our integer latent matrix for each attribute before encoding. For example, for L-dimensional latent attributes for N gaussians, we flatten the matrix to obtain a vector with L*N elements. This integer vector is then encoded using standard entropy coding approaches such as arithmetic coding [C]. These standard entropy coding approaches have been used in a large number of works [79, 81]. We will clarify and add an explanation in the revision.
>
> [C] Langdon, Glen G. "An introduction to arithmetic coding." IBM Journal of Research and Development 28.2 (1984): 135-149.
>
> ***
>
> ## Q: Confusing notations.
>
> `$\mathbf{l}_i$` is a generic symbol for the quantized integer latent for any of the four categories: rotation, scale, opacity and color.
> `$\mathbf{l}_{p_i}$` is the pre-gated residual for position attributes. $\alpha$ (without subscript, in bold font) is the set for all per-Gaussian alphas ($\alpha_i$). Each $\alpha_i$ corresponds to a $g_i$. We will improve our notation and add clarifications.
>
> Note: some symbols are not rendered correctly in the open review system, so we wrote it as plain code.
>
> ***
>
> ## Q: More detailed explanation on the challenges of per-frame training.
>
> Unlike offline reconstruction, per-frame training does not have access to future-frame information. This setup limits the capability to effectively reason about large scene changes (topological changes and highly varying appearance) [D]. Suppose an object suddenly appears or disappears, it is more difficult to (de-)allocate and update scene parameters to capture such changes. In the context of Gaussian splatting, it would be tricky to schedule densification and pruning of the Gaussians. We will add a more detailed explanation in the revision.
>
> [D] Richard A. Newcombe, Dense Visual SLAM, PhD Dissertation, 2012

---

### Official Review · Reviewer_jnXV · 2024-07-12

**Soundness:** 3
**Presentation:** 2
**Contribution:** 3
**Rating:** 6
**Confidence:** 2

**Summary:**

This paper introduces QUEEN, a framework designed to enable fast encoding (training) and decoding (rendering) for online free-viewpoint video (FVV) streaming using 3D-GS. Achieving high frame generation quality alongside real-time seamless streaming and rendering is challenging due to the intensive computation and large data volumes involved in this process. Towards this, QUEEN first proposes to apply all attribute residuals, unlike the state-of-the-art works that only use a subset, thereby maintaining high video quality. QUEEN also incorporates several components such as attribute residual quantization and position residual sparsity to accelerate the encoding and rendering process and reduce model size. Additionally, QUEEN leverages inter-frame redundancy, an important feature missed by previous works, to further enhance FVV generation efficiency. The evaluation demonstrates that QUEEN can reduce training time to only 5 seconds and achieve a rendering speed of around 350 FPS with a model size of merely 0.7 MB, all while maintaining near-optimal video quality.

**Strengths:**

+ The paper clearly discusses existing challenges in FVV, presents the proposed solutions logically, and details the evaluation results clearly.
+ The methodology is robust and the proposed components are thoroughly evaluated. The evaluation results demonstrate substantial improvements in model size and performance.

**Weaknesses:**

- Experimental Settings: The paper lacks details on some crucial experimental settings. For example, the number of bits used for attribute residual quantization is not specified. Similarly, for sparse position residuals, the precision is mentioned as full precision, does that mean fp32? For such case, could using bfloat16 (which reduces model size by half) solve the memory footprint issue without necessarily employing sparsity techniques?
- Evaluation Analysis: The analysis of evaluation results is insufficient. For instance, Table 2 (which is not referenced in the text), shows that introducing quantization and sparsity increases the quality (PSNR). Intuitively, these techniques should reduce quality, so an explanation is needed. Additionally, despite introducing extra processing steps (such as quantization, entropy encoding, etc.), training and rendering times do not increase. Detailed analysis/explanation on these evaluation results would be very beneficial.
- Perception of Quality: While numerical metrics like PSNR, SSIM, and LPIPS quantify the video quality, they may not fully reflect user perception of quality. A user study, though challenging, would significantly strengthen the paper.

**Questions:**

- Can you provide more details on the experimental settings for attribute residual quantization and sparse position residuals? Specifically, what number of bits is used for quantization, and is fp32 used for sparse position residuals?
- Can you explain why introducing quantization and sparsity improves quality (PSNR) and why the training and rendering times do not increase despite the additional processing steps?
- Is the receiver/rendering device the same as the encoding device? If the encoded frames/videos are sent to the client side (a different device), shouldn't the rendering be evaluated on a more reasonable device instead of the A100, which is server-level and too powerful for a typical client device?
- Can you discuss/study the accuracy-memory trade-off achieved by the quantization and sparsity techniques?

**Limitations:**

The authors acknowledge the limitation of exploiting inter-frame redundancy for very dynamic videos, which is reasonable. However, the paper evaluates video quality solely using metrics like PSNR and SSIM, which may not fully capture the viewing experience. Including a user experience study would provide a more comprehensive evaluation and strengthen the paper's conclusions. Also, including the accuracy-memory trade-off study with quantization and sparsity would be beneficial.

---

> ### Author Rebuttal · Authors · 2024-08-07
>
> Thank you for your thoughtful feedback. Please refer to our shared rebuttal (texts and PDF) for additional discussion and results. We address the specific questions below.
>
> ***
>
> ## Q: Experimental settings.
>
> - (a) The number of bits is dependent on the scene content as it relies on the amount of motion. This number can be fractional, on average, which is the standard for any entropy coding algorithm like arithmetic coding/Huffman coding. For example, for the Sear Steak scene in the N3VS datasets, on average, we require 0.68 bits for all the quantized attributes (corresponding to 0.5MB/frame). This depends on the entropy of the latents itself, which varies with each scene (Figure 7).
>
> - (b) For sparse position residuals, we use fp32 as full precision.
>
> - (c) For position residuals, we utilize sparsity as it has better compression ratios than fixed quantization like bfloat16, fp8/int8 and so on while still maintaining reconstruction quality (Table 2). As seen in Figure 5, only ~2% of gates are active, which corresponds to 98% sparsity and ~15x compression factor, (with additional storage costs for index locations) for the position residuals. In contrast bfloat16 would lead to only 2x reduction while still reducing the residual precision which is important for high reconstruction quality.
>
> ***
>
> ## Q: Evaluation analysis.
>
> Please see the detail discussion in the shared responses for all reviewers.
>
> ***
>
> ## Q: Perception of quality.
>
> We conduct an extensive A/B user study. For each vote, we show a pair of randomly chosen rendering results (from a test view that is not used in training) by our method and one of the baseline methods (3DStream and TeTriRF). We also show the ground truth video as a reference for the participants to make the decision. We ask the participant to choose the method that more faithfully matches the reference video. In total, we collected 285 responses from 15 participants within the timeline of the rebuttal. Tab. D2 summarizes the preference percentage of our method over the baseline methods. On both the N3DV and Google immersive datasets, the participants strongly prefer our results in comparison to the baseline methods.
>
> **Tab. D2**: User preference on visual results.
> | Baseline                |  Preference to Our Method (%)  |
> | :---------------------- | :-----------------------------: |
> | 3DGStream (N3DV)        |  76.67  |
> | 3DGStream (Immersive)   |  97.14  |
> | TeTriRF (N3DV)          |  96.67  |
>
> ***
>
>
> ## Q: Performance evaluation on a more reasonable client device.
>
> For fair comparison, we use the same A100 GPU for a consistent benchmark for measuring our rendering speeds (and training times) for our method as well as prior works such as 3DGStream [16]. Our rendering process is similar to the original work of 3D-GS [9]. The original paper of 3DGS and many recent follow-up work (e.g. [B]) demonstrate that Gaussian splatting rendering can work efficiently on a consumer-grade device (e.g., A6000 or RTX 2080). Due to the similarity of the rendering implementation, we believe our rendering speed performance would be at a similar level to what was shown in these 3DGS papers.
>
> [B] Xu et al. "Splatfacto-W: A Nerfstudio Implementation of Gaussian Splatting for Unconstrained Photo Collections." arXiv preprint arXiv:2407.12306 (2024).
>
> ***
>
> ## Q: Accuracy-memory trade-off.
>
> Please see the shared rebuttal response.

---

### Author Rebuttal · Authors · 2024-08-07

We thank the reviewers for the insightful comments and for acknowledging the novelty of our design (jnXV, 9x86), a “solid submission” (L5E8), superior performance (all 4 reviewers) and
extensive evaluations (all 4 reviewers). We address the common issues in this **shared response** and we will address **individual questions** to each reviewer. We also submitted **a PDF** that contains further quantitative results (Fig. R1, Fig. R2, and Tab. R1).

***

## Q: Evaluation analysis and confusing Tab. 2 (jnXV and 9x86).

We thank the reviewers for pointing it out and we will address this further in the revision.

- (a) We found that quantizing the scaling attribute leads to stable optimization and better-behaving Gaussians. Fig. R2 visualizes the histogram of the scale attributes (on log scale) with and without quantization at different frame instances for an N3DV scene. We see that quantizing the scale attribute leads to stable histograms without a continuous increase in the size of the Gaussians (note that size here refers to the actual size of the Gaussians measured by the scaling attribute and not the storage size post-training). In contrast, the baseline training at full-precision leads to growing Gaussian sizes (red box) which leads to unstable optimization (A similar result for the opacity attribute is observed in [84] where quantization acts as a regularizer leading to better optimization with fewer outlier gradients). Larger Gaussians also lead to slower rendering speeds along with slow training times. This is because rasterizing large Gaussians across multiple tiles can lead to more GPU memory transfers from DRAM to SRAM which can slow down the rendering speeds.

- (b) Thus, while the quantization and sparsity frameworks introduce overhead in training times and rendering speed, quantizing the scaling attribute has the opposite effect of improving training times due to faster rendering. To support this, we show quantitative results in Tab. D1 with and without scaling quantization (SQ) while compressing other attributes on the N3DV dataset. Quantizing scaling attributes improves reconstruction quality while still reducing memory and training time. This largely fits what we observe in Tab. 2.

**Tab. D1**: scaling quantization (SQ) improves quality, compression rate and training time.

| Config.              |  PSNR  | Size (MB) | Training Time (sec)  |
| :------------------- | :-------: | :----: | :----: |
| w/o SQ                |  31.69   | 4.39|  11.01 |
| w/ SQ.                 |   **32.08**   | **0.69** |  **7.07**  |

- (c) Different trends of training/rendering speeds between N3DV and Immersive datasets: the Immersive dataset contains more challenging scene dynamics (e.g., a person entering and leaving the scene). To capture this highly dynamic content, we schedule a more aggressive Gaussian densification on the Immersive dataset than the N3DV dataset. This strategy leads to a higher increase in the number of Gaussians. We think these differences lead to the different trends in training/rendering speeds.

***

## Q: Accuracy-memory trade-off and ablation study on quantization setup (Reviewer jnXV and 9x86).

- (a) We show the PSNR-size and PSNR-training-time tradeoffs in Fig. 6 (Appendix A) by varying the number of training iterations for both quantization and sparsity techniques on position attributes. More iterations correspond to longer training times, which result in higher storage size due to more non-zero residuals and higher entropy. However we observe the quality (PSNR) reaches a plateau after certain iterations.

- (b) A knob for controlling sparsity (and thereby memory costs) is the $\lambda_{reg}$ loss coefficient (Eq. 11) which allows for increasing/decreasing amount of sparsity based on the strength of the regularization. We visualize this in Fig. R1 (b) where increasing the lambda_reg coefficient leads to higher sparsity/lower memory and lower reconstruction quality.

- (c) While $\lambda_{reg}$ controls the sparsity of the position residuals to obtain accuracy-memory tradeoffs, we further experiment with a different knob for controlling the tradeoffs pertaining to the quantization framework. We experiment with a regularization loss to reduce the entropy of the latents as lower entropy corresponds to lower memory (but also lower reconstruction quality). While this can be done via learnable probability models as in [81], it leads to higher training costs in terms of time and memory. We instead observe that the probability distribution of the various attribute residuals at each time-step is unimodal and is close to a laplacian/Gaussian distribution. As a unimodal distribution has entropy proportional to the variance [A], we enforce a loss on the standard deviation of the latents with a tradeoff parameter $\lambda_{std}$ controlling the effect of this regularization loss. Fig. R1 (a) in the rebuttal shows results on the N3DV dataset by varying $\lambda_{std}$. We observe that increasing the parameter reduces the entropy costs leading to lower memory costs, but lower reconstruction quality, and vice versa.

- (d) Effect of latent dimension: We provide additional analysis on the effect of latent dimension for the various attributes in Table R1 (rebuttal PDF). In general, latent dimension does not have a significant effect on reconstruction quality or model size. The vector quantization is a flexible scheme to decrease the per-dimension entropy. After the end-to-end training, we apply entropy encoding. This will further exploit the redundancy and reduce the impact of latent dimension. A better variable/knob to achieve the tradeoff between quality-memory or quality-time in our framework is the entropy loss/variance coefficient or the total number of iterations as mentioned above.

[A] Chung et al. "Bounds on variance for unimodal distributions." IEEE Trans. on Information Theory 63.11 (2017): 6936-6949.

---

### Comment · Area_Chair_6sVo · 2024-08-09

Dear reviewers, do the authors' responses answer your questions or address your concerns? Thanks.

---

> ### Comment · Area_Chair_6sVo · 2024-08-12
>
> Dear reviewers, as we approach the final two days, please take a moment to review the author's responses and join the discussion. Thank you!

---

> > ### Comment · Area_Chair_6sVo · 2024-08-14
> >
> > Dear reviewers, the authors are eagerly awaiting your response. The author-reviewer discussion closes on Aug 13 at 11:59 pm AoE. Thanks!

---

### Decision · Program_Chairs · 2024-09-25

**Decision:**

Accept (poster)

**Comment:**

The paper introduces QUEEN, a framework for efficient streaming of free-viewpoint videos using 3D Gaussian Splatting. It focuses on high-quality reconstruction, fast training, and reduced model size. QUEEN outperforms state-of-the-art methods on various benchmarks, particularly in dynamic scenes.
We are glad to accept the paper, based on the reviewers' comments which are all positive.
We encourage the authors to include the suggested revisions by the reviewers in their camera-ready version (e.g., clarify the quantization process and experimental details, provide a more detailed analysis of the results, address hyperparameter sensitivity, ensure code release for reproducibility, and discuss limitations and future directions).